# Revisiting the Last Ice Area projections from a high-resolution Global Earth System Model
Madeleine Fol [1] ✉, Bruno Tremblay[1,2], Stephanie Pfirman[3], Robert Newton[2], Stephen Howell [4] & Jean-François Lemieux [5]

The Last Ice Area—located to the north of Greenland and the northern Canadian Arctic Archipelago—is expected to persist as the central Arctic Ocean becomes seasonally ice-free within a few decades. Projections of the Last Ice Area, however, have come from relatively low resolution Global Climate Models that do not resolve sea ice export through the waterways of the Canadian Arctic Archipelago and Nares Strait. Here we revisit Last Ice Area projections using high-resolution numerical simulations from the Community Earth System Model, which resolves these narrow waterways. Under a high-end forcing scenario, the sea ice of the Last Ice Area thins and becomes more mobile, resulting in a large export southward. Under this potentially worst-case scenario, sea ice of the Last Ice Area could disappear a little more than one decade after the central Arctic Ocean has reached seasonally ice-free conditions. This loss would have profound impacts on ice-obligate species.

Within a few decades, most of the central Arctic Ocean is expected to become seasonally ice-free. The exception to this trend is the region surrounding the Queen Elizabeth Islands (QEI) and areas north of the Canadian Arctic Archipelago (CAA) and Greenland[1,2], where some of the oldest and thickest sea ice accumulates. Here, a perennial sea ice cover is expected to persist for some time period even after the remainder of the central Arctic Ocean is ice-free in summer. This region is termed the Last Ice Area (LIA— see red contour on Fig. 1) concept that was first introduced at a press conference at the American Geophysical Union Annual meeting[3,4] and served as the basis for the LIA flagship project of the World Wildlife Fund— Canada (wwf.ca). In 2019, this led to the interim creation of the Tuvaijuittuq Marine Protected Area within the LIA designated by Canadian ministerial order (Tuvaijuittuq means "the ice never melts" in Inuktitut). The LIA also encompasses the Remnant Arctic Multi-Year Sea Ice and the Northeast Water Polynya ecoregion, which is proposed as a potential UNESCO World Heritage site[5]. The stability of this region is crucial for preserving the Arctic ecology as it provides a suitable habitat for ice-dependent and ice-obligate species, including polar bears, belugas, bowhead whales, walruses, ringed seals, bearded seals, and ivory gulls[6,7]. In August 2024, interim protected status was extended to the Tuvaijuittuq Marine Protected Area within the LIA for up to 5 years "while the Government of Canada works with partners to consider long-term protection" (www.dfo-mpa.qc.ca). The results presented in this study provide foundational information for understanding future sea ice conditions in this area.

Up until now, Global Climate Models have provided valuable insights into the thermodynamics and large-scale dynamic processes responsible for the presence and future evolution of the LIA, but lacked the spatial resolution needed to resolve the narrow waterways of the CAA and Nares Strait[3,8,9], key outlets for ice exported out of the LIA[10–12]. Consequently, earlier LIA projections are biased by models that did not account for transport through these channels. This limitation was less critical in the past, when sea ice was mostly landlocked in the QEI and transport was blocked by ice arches across Nares Strait[12–14]. However, with the recent increase in the export of pan-Arctic sea ice through the CAA and Nares Strait[10,11,15] there is a need to revisit the LIA projections now considering sea ice transport through these gateways.

The existence of the LIA is primarily due to sea ice dynamic processes, specifically the convergence of sea ice on the northern shores of Canada and Greenland, driven by the large-scale circulation of the Beaufort Gyre and the Transpolar Drift Stream. Most sea ice in the LIA forms in marginal seas - principally Siberian shelf seas - and the central Arctic[9,16]. Once in the LIA, sea ice may (1) be incorporated in the southern branch of the Beaufort Gyre and reinjected into the central Arctic, (2) flow southward into the QEI through the southern CAA and the shipping lanes of the Northwest Passage,

[1]Department of Atmospheric and Oceanic Sciences, McGill University, Montréal, QC, Canada. [2]Lamont-Doherty Earth Observatory, Columbia University, Palisades, NY, USA. [3]College of Global Futures, Arizona State University, Tempe, AZ, USA. [4]Climate Research Division, Environment and Climate Change Canada, Toronto, ON, Canada. [5]Recherche en Prévision Numérique Environnementale/Environnement et Changement Climatique Canada, Dorval, QC, Canada. ✉e-mail: madeleine.fol@mail.mcgill.ca

**Fig. 1 | Map of the Last Ice Area (red contour) as defined by the World Wildlife Fund.** The map includes the Queen Elizabeth Islands (QEI, red), the region north of the Canadian Arctic Archipelago (LIA-N, green) and part of the CAA-South (yellow), with major gates connecting the LIA-N, the QEI, the CAA-S (orange) and Northern Baffin Bay (white). These include the QEI-In gates [Ballantyne Strait (Ball.), Wilkins Strait, Prince Gustaf Adolf Sea (Pr. G.A.), Peary Channel, Sverdrup Channel (Sv.), and Eureka Sound], the QEI-Out gates [Fitzwilliam Strait (Fitz.), Byam Martin Channel (Byam M.), Penny Strait, Cardigan Strait and Hell Gate (Hell)], Fram Strait, Nares Strait, Jones Sound, Lancaster Sound, Amundsen Gulf (Amund.) and M'Clure Strait. The Tuvaijuittuq Marine Protected Area (in the LIA-N) is hashed and the Remnant Arctic Multi-Year Sea Ice and the Northeast Water Polynya ecoregion is dotted.

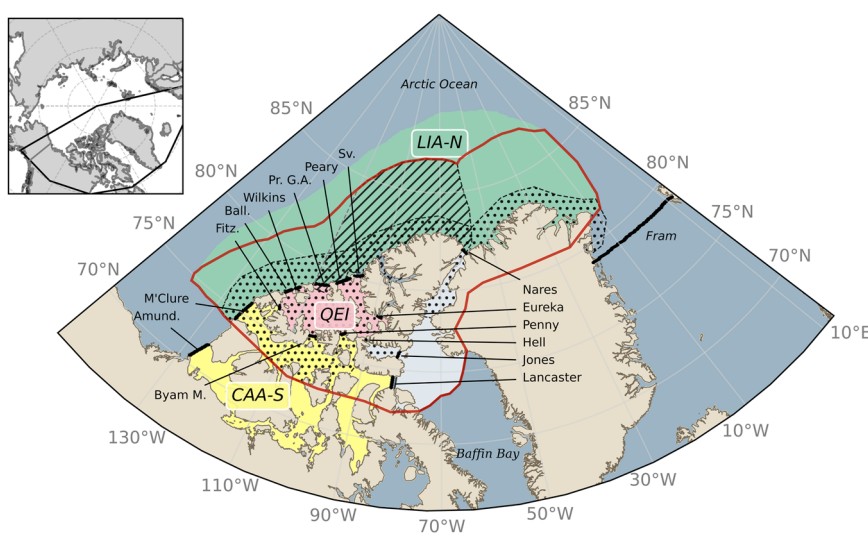

(3) flow through Nares Strait towards Baffin Bay or (4) flow through the Fram Strait towards the North-Atlantic Ocean (Fig. 1). Once exported out of the central Arctic Ocean, sea ice drifts southward in the southern CAA, Labrador and Greenland seas where it melts. These characteristics of sea ice circulation are only partly resolved in low resolution models (1 and 4). This paper examines the impacts of sea ice transport through the QEI and Nares Strait (2 and 3) on projections of the LIA, while also addressing larger-scale processes.

The QEI are characterized by high concentrations of multi-year ice that extend into and across the Parry Channel and waterways of the Northwest Passage of the southern CAA[17–20]. The convergence of sea ice in the LIA creates shore-aligned ridges with a mean thickness of 4–6 m[21], and that can extend up to 25 m[17,22]. Consequently, sea ice within the CAA contains a mix of multi-year ice and locally-grown 1st-year ice that mostly remains landfast from November to July[18]. Arch-like ice features often form at the entrance of narrow passages when converging sea ice builds up into ridges due to the pressure exerted by drifting ice against landmasses. Ice arches have previously formed at the entry gates of the QEI and in Nares Strait, preventing southward outflow for several months each year of the thick multi-year ice from the Arctic Ocean through these channels. Since 1997, there has been a significant increase in the amount of multi-year ice imported into these regions associated with a reduced landfast ice arch duration[11,12]. Moreover, the LIA has lost sea ice volume at a rate twice as large as the central Arctic Ocean[12]. This creates a weaker ice cover that is more prone to break-up events during periods of high winds, faster drift, and increased open water areas allowing the ice albedo feedback to operate more effectively[23]. Also, the departure from free-drift conditions is a function (to the first order) of the ice-ice interactions (i.e., the rheology), which decreases exponentially with decreasing sea ice concentration[24]. This increased mobility of sea ice heightened the transport through gates of the QEI by 10,000 km² per decade (1997–2018)[11] and led to events of early collapse or absence of ice arches in Nares Strait, doubling the average export from the 1997–2009 period[10,12,25].

Therefore, in this study, we revisit the LIA projections using a high-resolution (0.1°) version of the Community Earth System Model (CESM1.3-HR[26,27]) that resolves ice export through the CAA and Nares Strait. The boundaries of the LIA vary in the literature depending on the long-lasting sea ice cover contour in model projections[9,16]. We adopt the definition provided by the WWF which includes the LIA-North (LIA-N), the QEI, and northern Baffin Bay (see red contour on Fig. 1). We divide the LIA into three main regions: the LIA-N (1.14 million km²), the QEI (0.16 million km²), and the southern CAA (CAA-S-0.59 million km²—Fig. 1). Although the WWF definition of the LIA includes sea ice in northern Baffin Bay and Jones Sound, these areas are currently seasonally ice-free and do not contribute to

the LIA (Fig. 1). To this end, we compute sea ice area (SIA) fluxes through entry and exit gates of the QEI, the CAA-S and Nares Strait (Fig. 1).

First, we evaluate the pan-Arctic ice that feeds the LIA and assess regional biases in comparison to lower-resolution global climate models, specifically CESM1.3-LR and CESM2-LE, the Pan-Arctic Ice Ocean Modeling and Assimilation System (PIOMAS), and observed sea ice extent, area, thickness, and fluxes derived from high-resolution satellite imagery. Then, we present and interpret results for sea ice within the LIA in the context of the model biases. Finally, we discuss the dynamic and thermodynamic contributions, and potential feedback, responsible for the disappearance— i.e., near zero SIA—of perennial sea ice of the LIA.

The model forcing fields for CESM1.3-HR are derived from the RCP8.5 emissions scenario, the highest-emission pathway used in IPCC projections of future warming (90th percentile level, representing a temperature increase of about 4.3 °C by 2100, relative to pre-industrial temperatures) and include no mitigation relative to historical emissions growth rates. We present results from the high-end forcing scenario (1) because it is the scenario of choice for most currently available high-resolution global climate model simulations worldwide (e.g., equivalent forcing scenarios for GFDL CM2.6[28] and HighResMIP CMIP6[29]), and (2) because it provides potentially a worst-case scenario. CESM1.3-HR shows significant negative biases in sea ice extent[26], area, and thickness when compared to observations and lower-resolution models (see Section "Comparison of simulated and observed historical sea ice conditions (1920–2023)"). These biases are likely linked to a warmer sea surface temperature, which may result from the explicit parameterization of upward ocean heat transport by mesoscale and sub-mesoscale eddies[26]. This positive bias in the vertical temperature profile leads to a stronger meridional heat transport, a 25% stronger Arctic amplification, and consequently an underestimation of the sea ice extent[26], area, and thickness in the Northern Hemisphere. Therefore, considering these negative biases and the RCP8.5 warming scenario, the results presented here should be viewed as a worst-case scenario for LIA ice loss.

Previous studies using climate models have dedicated considerable effort into estimating a realistic timing of a seasonally ice-free central Arctic, by trying to reduce major uncertainties rising from the model, climate variability and the uncertainty in future greenhouse gas scenarios. Most CMIP3 and CMIP5 models failed to reach central Arctic ice-free conditions at the end of the century, in apparent contradiction to the observed rapid decline of central Arctic sea ice[30,31]. The CMIP6 generation of models showed improvements in reproducing observed sea ice conditions[1,2]. CMIP6 models predict that the central Arctic could experience its first ice-free summer as early as the 2020 s or 2030 s, with a high likelihood of this occurring by 2050. Under all warming scenarios, the central Arctic Ocean is

expected to be ice-free in most summers, including natural variability, by mid-century, between 2035 and 2067[1]. Only under the lowest warming scenario (SSP1-1.9) do CMIP6 models keep significant thick September sea ice north of the CAA at the end of the century[1]. In moderate to high warming scenarios, the thick sea ice of the QEI disappears thermodynamically 10–20 years after reaching a consistent seasonally ice-free central Arctic[1]. As part of the CMIP6 suite, HighResMIP models that resolve the channels of the CAA and Nares Strait generally overestimate SIA loss in the LIA compared to observations, projecting both regional and pan-Arctic seasonally ice-free conditions by around 2050[32].

These differences between CMIP3, CMIP5, and CMIP6 show that uncertainties regarding projections for regional ice-free conditions using CMIP models are greater than those for pan-Arctic projections and are highly sensitive to the specific models employed[33]. Therefore, rather than attempting to project the exact timing of the regional disappearance of the LIA, we provide an estimate of the timescale required to drain the LIA once the central Arctic Ocean becomes seasonally ice-free, thereby accounting for the biases and uncertainties in CESM1.3-HR.

## Results

### Comparison of simulated and observed historical sea ice conditions (1920–2023)

The rate of decline and interannual variability in September sea ice extent, SIA, and May mean ice thickness simulated by the CESM1.3-HR are in general agreement with observations and PIOMAS. This agreement is in the face of negative biases of ~3 million $km^2$ in sea ice extent and SIA (similar to that of CESM2-LE), and ~0.4 m in mean thickness (Figs. 2, 3, and 4) associated with a bias towards warmer sea surface temperature compared with observations[26]. The larger negative bias in CESM1.3-HR SIA (compared to sea ice extent) is mainly the result of a broader marginal ice zone compared with observations, covering the full Arctic Ocean as early as 2001-2020 (Fig. S1). The Arctic-wide thinner ice cover feeding the LIA leads to a somewhat earlier onset of the decline in September sea ice extent and August-September-October mean sea ice concentration in the LIA-N and the QEI simulated by CESM1.3-HR compared with observations (Figs. 2 and S2). In the CAA-S, the rate of decline of the September sea ice extent agrees with observations (akin to that of CESM2-LE), despite a negative bias of ~0.1 million $km^2$ (Fig. 2). Note that the landmasks of the low-resolution models (CESM1.3-LR and CESM2-LE) are larger by ~22% and ~15% for the CAA-S and the QEI respectively, leading to a possible negative bias in the simulated September sea ice extent compared with CESM1.3-HR and observations.

The ice thickness distribution has a bias towards thinner sea ice when compared to PIOMAS with an overestimation in the mid to large ice thickness categories (1.39–2.47 m), in the LIA-N, QEI and CAA-S (Figs. 3 and S1). Note that PIOMAS itself has a bias towards thinner sea ice north of the CAA compared with observations from ICESat, airborne-electromagnetic induction and submarine-based upward looking sonar measurements[33].

The simulated August-September-October mean sea ice concentration in the LIA-N and the QEI has mean negative biases of 20% and 9% respectively, compared with CDR (Fig. S2). While there are negative biases in sea ice extent, SIA, and sea ice thickness in CESM1.3-HR, the sea ice concentration in the LIA-N is similar to that of CESM1.3-LR, because of the omnipresent sea ice convergence against the northern CAA coastline. Since sea ice resistance is mostly dependent on ice concentration[24], thicker sea ice would lead to similar SIA fluxes in the LIA.

The simulated seasonality of SIA fluxes is in excellent agreement with observations in the Fram Strait and Amundsen Gulf for the historical period (Fig. 5). At other gates, notable differences include the reverse seasonality in Nares Strait with a positive bias in winter and negative bias in the fall (due to weaker ice sea ice barriers in the model), earlier onset of fluxes in the QEI (May compared to August, when the ice-albedo feedback is more active), and the systematic overestimation of sea ice fluxes in QEI-in, QEI-out, M'Clure, and Lancaster Sound by 5–20 × $10^3$ $km^2$. These biases add up to annual simulated SIA fluxes larger

by a factor of approximately two compared with observations. The positive bias at the entry and exit gates of the QEI, results in an annual SIA divergence of 21 × $10^3$ $km^2$ (($113 \pm 10 - 92 \pm 14$) × $10^3$ $km^2$), compared with an observed convergence of 9 × $10^3$ $km^2$ (($28 - 37$) × $10^3$ $km^2$) for the same time period (2017–2021, Fig. 6). Biases at the gates of the CAA-S (in and out) lead to a 2017–2021 mean annual simulated divergence of 164 × $10^3$ $km^2$ (($360 \pm 71 - 196 \pm 48$) × $10^3$ $km^2$), which is very similar to the observed mean divergence of 159 × $10^3$ $km^2$ (($200 - 41$) × $10^3$ $km^2$, Fig. S3).

### Projections of the LIA

Within the context of this high-end forcing scenario, the central Arctic Ocean becomes seasonally ice-free in 2020 for the first time, and then continuously from 2035 on (Fig. 2). Due to the pan-Arctic biases in sea ice extent and thickness in CESM1.3-HR, these projections fall within the early range of seasonally ice-free pan-Arctic estimates from CMIP6 models[1]. Regionally, the probability density functions of the May ice thickness in the LIA-N, QEI and CAA-S narrow with time, with a shift in the peak toward thinner sea ice (Fig. 3). In addition, the LIA-N and the QEI transition from (small) dynamically driven to (large) thermodynamically driven SIA losses is mostly responsible for the significant reduction in September SIA in the same two regions (Fig. 4 and S4). Basal melt dominates over surface melt in the LIA-N, and is of equal importance in the QEI and CAA-S (Fig. S5). The LIA-N and the QEI are regions of thicker sea ice, and therefore show smaller area loss for a reduced ice thickness compared to the CAA-S. In the CAA-S, SIA loss remains thermodynamically driven until near the end of the century. During the 2040–2080 time period, the LIA-N is mostly seasonally ice-free with a predominance of the 0.64–2.47 m category in May ice thickness distribution (Fig. 3). The loss of SIA is thermodynamically driven in the QEI (Figs. 4 and S4), with no net dynamic loss (flux in = flux out, Fig. 6), but with a large increase in the flux throughflow magnitude serving to drain the seasonal thin cover and the remnant thick ice of the LIA-N (Fig. 3). During this thermodynamically driven SIA loss period (2040-2090), the model shows significant negative correlation coefficients ($r \leq -0.82$, $p < 0.001$) between the melt season's integrated thermodynamic and dynamic SIA loss in all regions (Fig. S6). This strong negative feedback is a direct result of a reduced residence time of sea ice in all three regions: LIA-N, QEI and CAA-S.

As the climate warms, the peak in the seasonal SIA fluxes through the QEI and Nares Strait increases in magnitude and occurs earlier in the melt season (Fig. 5). This trend persists until 2080–2100, when winter SIA declines. This decline results in less landfast ice which allows newly formed sea ice in the winter to be exported out of the Arctic Ocean through the CAA and Nares Strait, leaving little to no ice to be exported in the summer. The seasonality in SIA fluxes at the gates of the CAA-S shows less variability, with a gradual increase during winter and a decrease during summer. In Fram Strait, SIA fluxes have no apparent shift in seasonality but largely decrease due to the reduced availability of sea ice and the retreat of ice in the LIA-N. This suggests that at this time, the LIA SIA loss through transport, historically dominated by Fram Strait, will shift to QEI and Nares Strait, as suggested in recent studies[11,12,15,25]. The annual SIA exports through the QEI-in, QEI-out, and Nares Strait gates increase by factors of 5.1, 2.0 and 1.4 respectively, from the 1920–1980 period to the 2040–2080 period (Fig. 6). The CAA-S annual SIA budget shows greater stability, with a decrease in net divergence, primarily resulting from the increased fluxes through gates of the QEI (Fig. S3).

From 2020 to 2080, the August-September-October mean sea ice concentration in the QEI drops well below 90%, leading to low ice-ice interactions (Fig. S2). In the LIA-N, the trend in August-September-October mean sea ice concentration reverses around 2035, when only a few high sea ice concentration grid cells remain (Figs. S1–2). With sea ice concentration in the LIA-N remaining between 40% and 90%, which allows year-long ice intake into the QEI, we argue that the simulated sea ice velocities at QEI gates are in near free-drift during the 2040–2080 period. This regime of ice transport is reached when ice-ice interactions are small and their magnitude depends chiefly on the simulated wind forcings of the

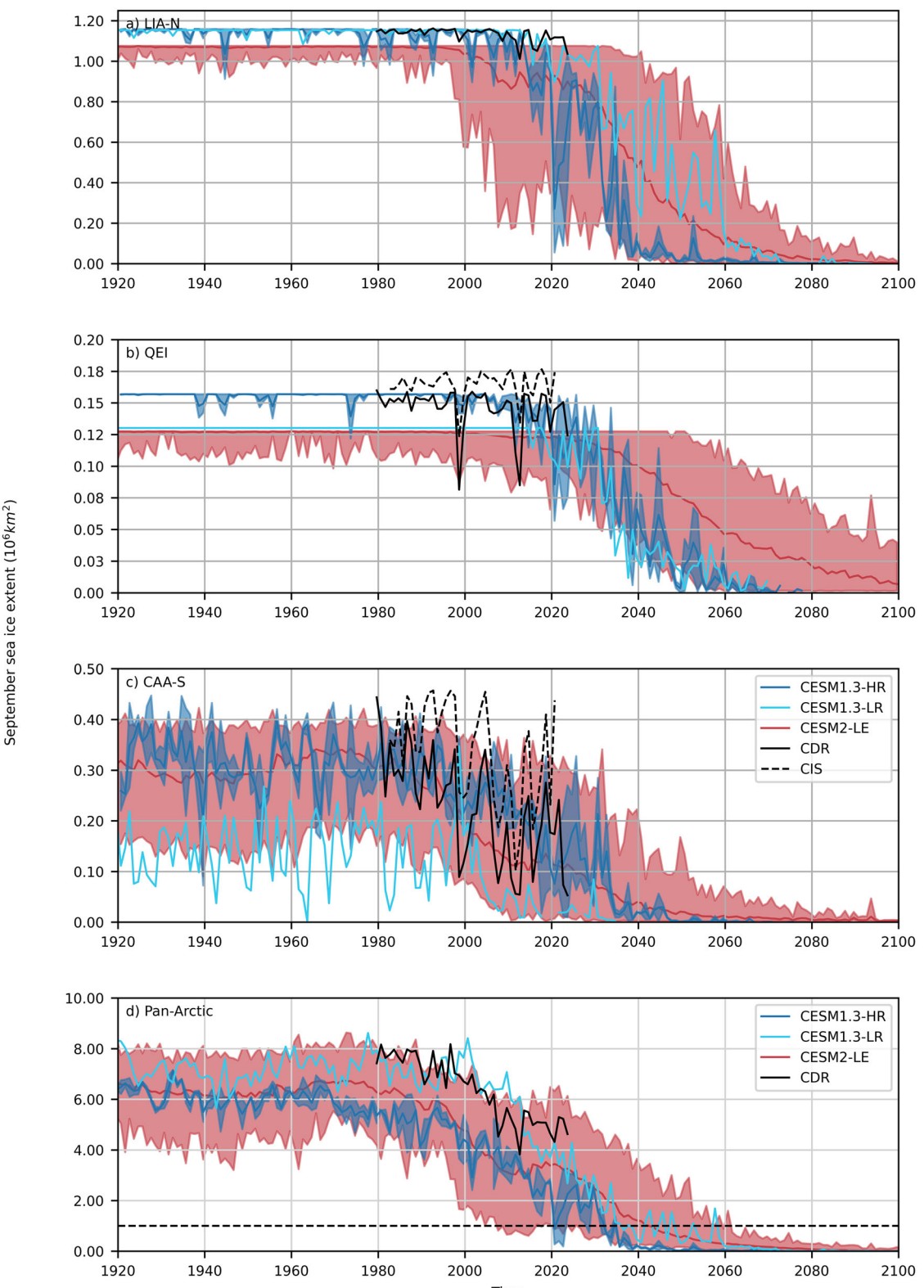

**Fig. 2 | Observed and simulated September sea ice extent.** Observed and simulated ensemble mean and range September sea ice extent for CESM1.3-HR (dark blue), CESM1.3-LR (light blue), CESM2-LE (red), CDR (black-full lines), and CIS (black-dashed) for the LIA-N (**a**), QEI (**b**), CAA-S (**c**), and the pan-Arctic Ocean (**d**). The black dashed line is the 1 million km² ice-free Arctic as defined by the Intergovernmental Panel on Climate Change. Note that the vertical scale of each panel is different.

model. As Arctic wind speeds are projected to increase by 6.4–9.6% by the end of the century[34], surface wind stress is expected to rise by ~13–20%.

During the transition to a seasonally ice-free central Arctic, the CAA plays a crucial role in the stability of the LIA, continuing to prevent outflow

of sea ice from the LIA-N and originating from the central Arctic Ocean. The CAA-S exhibits smaller changes in its May ice thickness distribution, melt season length, SIA fluxes, and thermodynamic processes continue to be dominant, in contrast with the LIA-N and QEI (Figs. 3, 4, S3, S4). This

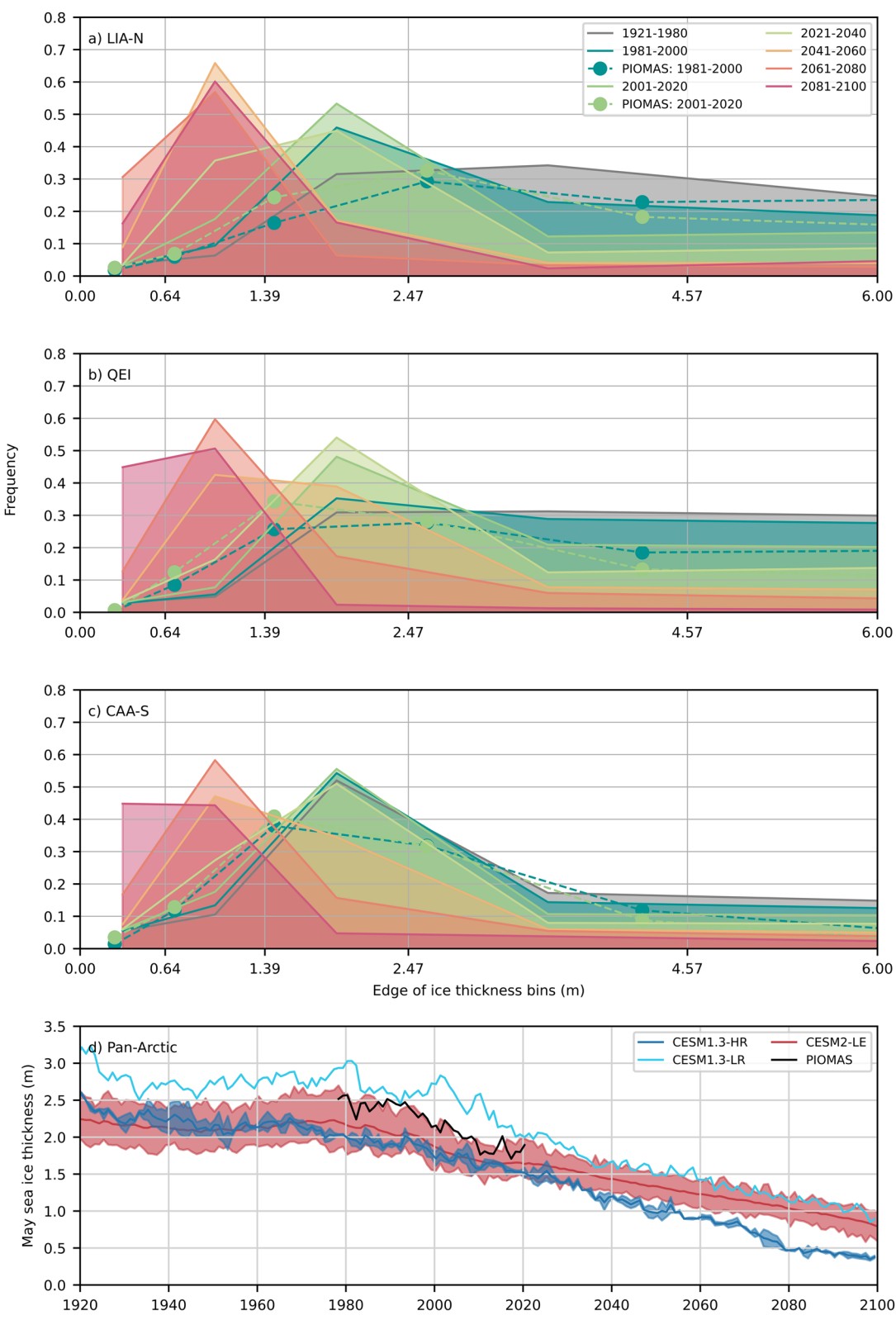

**Fig. 3 | Simulated May ice thickness.** Simulated ensemble mean (CESM1.3-HR-full lines—PIOMAS-dashed) 20-years mean May ice thickness distribution from 1921 to 2100 for the LIA-N (**a**), QEI (**b**), and CAA-S (**c**). Pan-Arctic simulated mean and range mean May sea ice thickness (**d**) from CESM1.3-HR (dark blue), CESM1.3-LR (light blue), CESM2-LE (red), and PIOMAS (black).

stability in the CAA-S suggests a limited (but possibly sustainable) drainage of the LIA through the QEI, and allowing sporadic replenishment of the QEI with thick multi-year ice from the north. This is in agreement with the CAA's drain-trap mechanism[35,36], where a gradual and sustained melt of

multi-year ice in the southern CAA, opens the door for a rapid replenishment of multi-year ice from the central Arctic Ocean into the QEI and the southern CAA[17,19,37]. Spectral analysis of the detrended thermodynamic, dynamic (advection + ridging), and flux-derived dynamic SIA loss reveal a

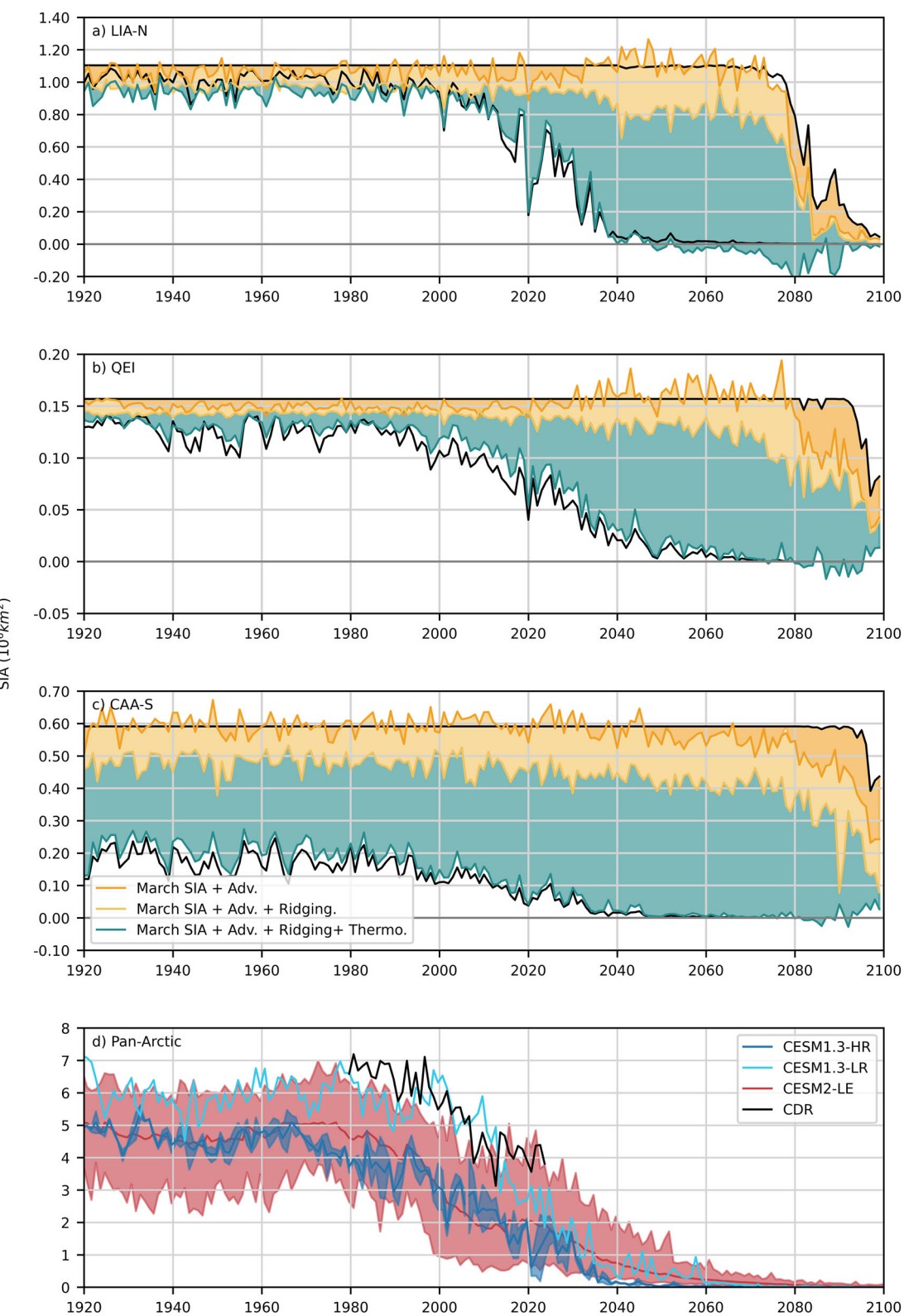

**Fig. 4 | Simulated March and September SIA with dynamic and thermodynamic SIA loss.** Simulated ensemble mean March and September SIA (black) together with dynamic (advection and ridging) and thermodynamic SIA loss integrated spatially and temporally over the melt season for the LIA-N (**a**), QEI (**b**), and CAA-S (**c**). Note that the difference between the integrated dynamic and thermodynamic area loss (orange + yellow + blue) and the two solid black lines is a measure of the error in the area budget (see Online "Methods" section). Pan-Arctic simulated and observed mean and range mean September SIA (**d**) from CESM1.3-HR (dark blue), CESM1.3-LR (light blue), CESM2-LE (red), and CRD (black). Note that the vertical scale of each panel is different.

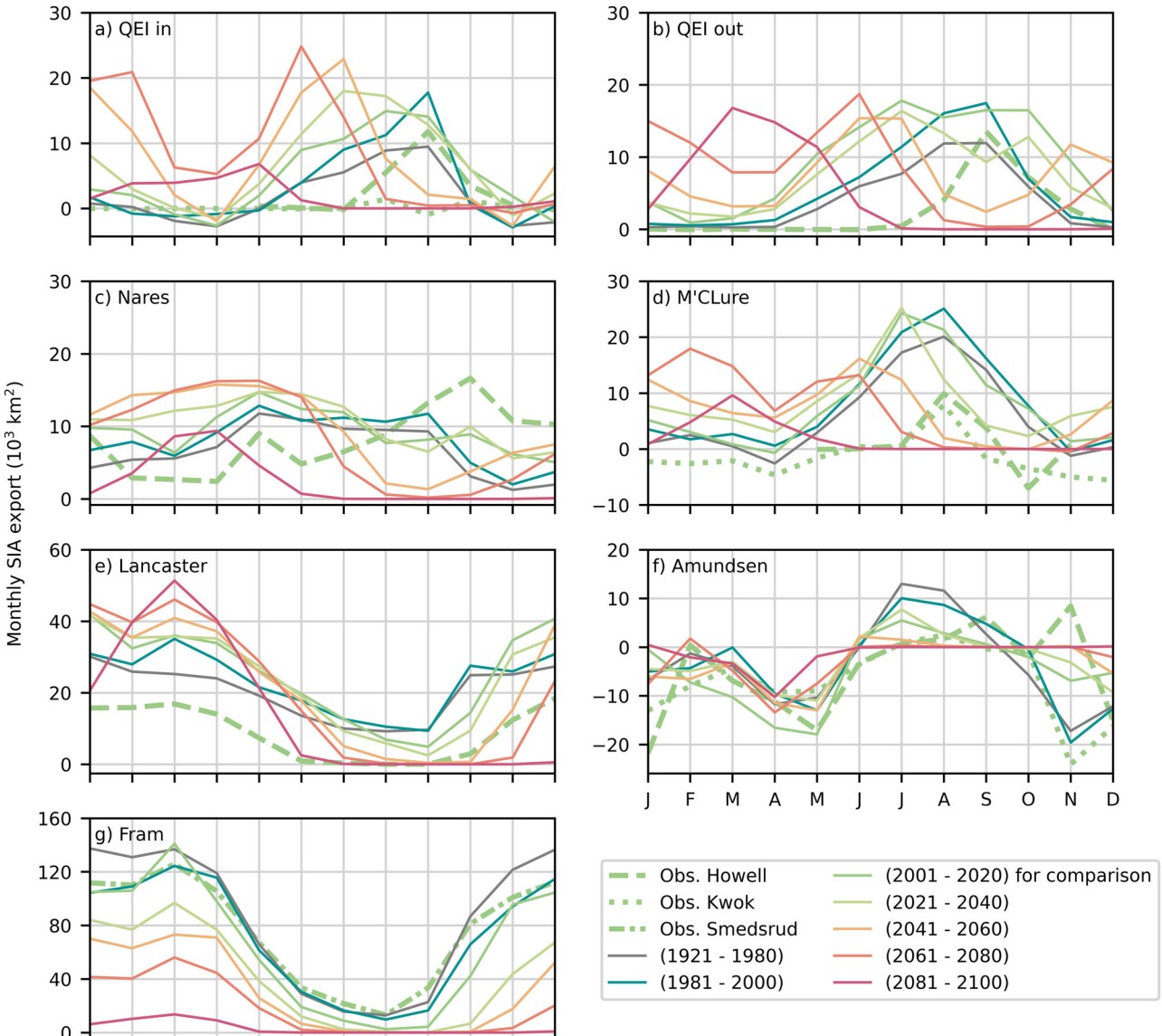

**Fig. 5 | Simulated seasonal cycle of SIA export through gates of the CAA and Nares and Fram straits.** Simulated (full lines) 20-years averaged seasonal cycle of SIA export through the QEI-In (**a**), QEI-Out (**b**), Nares Strait (**c**), M'Clure Strait (**d**), Lancaster Sound (**e**), Amundsen Gulf (**f**) and Fram Strait (**g**) from 1921 to 2100, except for the 1921–1980 long time average, and the 2001–2020 period where subsets of years are selected to coincide with observational records. The observed (and comparable simulated) mean seasonal cycles are calculated over 2001–2020 for the QEI-In and M'Clure Strait, 2017–2021 for the QEI-Out, Nares Strait, Lancaster Sound, and Amundsen Gulf, and 2000–2014 for the Fram Strait.

dynamic lagged response between the LIA-N, the QEI, and the CAA-S, with periods of ~4–6 years (Fig. S7). A significant coupled thermodynamic SIA loss cycle occurring in the CAA-S every 6 years indicates dominant melting events, allowing for increased sea ice transport in the subsequent years reminiscent of the drain-trap mechanism.

## Discussion

As defined by the Intergovernmental Panel on Climate Change, the Arctic is referred to as seasonally ice-free when only 1.0 million km² of sea ice remain. Earlier studies from low resolution climate models (CCSM or CESM)[3,8,9] projected that sea ice cover will continue to persist in the LIA several decades past this time. Our results, under the RCP8.5 forcing scenario, indicate that the LIA has little chance of persisting once central Arctic sea ice is gone, given that the thick sea ice of the LIA is gradually lost through melting and export via the QEI, Nares, and Fram straits—as recent events suggest[10,12,15]. Under this high-end scenario, as the Arctic

Ocean transitions to a seasonal sea ice cover, the reservoirs of thick multi-year ice in the LIA-N and QEI gradually thin until 2040–2060 with a peak in ice thickness distribution under 1.39 m (Fig. 3). This gradual loss of the thick ice within the LIA operates through a slow increase in sea ice advection through the QEI and Nares Strait and out of the LIA-N from 2000 until 2040, a period during which the length of the melt season increases, the ice arches are weaker, and SIA loss becomes thermo-dynamically driven. At this point, sea ice in the thicker ice categories is reduced by a factor of two in the LIA-N and the QEI, leading to nearly continuous (unobstructed) dynamic export of Arctic sea ice through the QEI and Nares Strait from 2040 to 2080. This, coupled with enhanced melting, and a reduction in sea ice availability in the source regions (i.e., the LIA-N and central Arctic—Figs. 2, 4), results in the complete drainage of the LIA (Fig. 2—see below for an estimate based on SIA fluxes). Finally, SIA fluxes through the QEI reduce during the 2081–2100 period, as winter sea ice declines in the LIA (Fig. 4) and the central Arctic.

**Fig. 6 | Observed (hashed bars) and simulated (full bars) annual ensemble mean SIA fluxes in the QEI and Nares Strait.** Positive fluxes represent SIA entering the CAA and negative fluxes exiting the CAA or the Arctic Ocean.

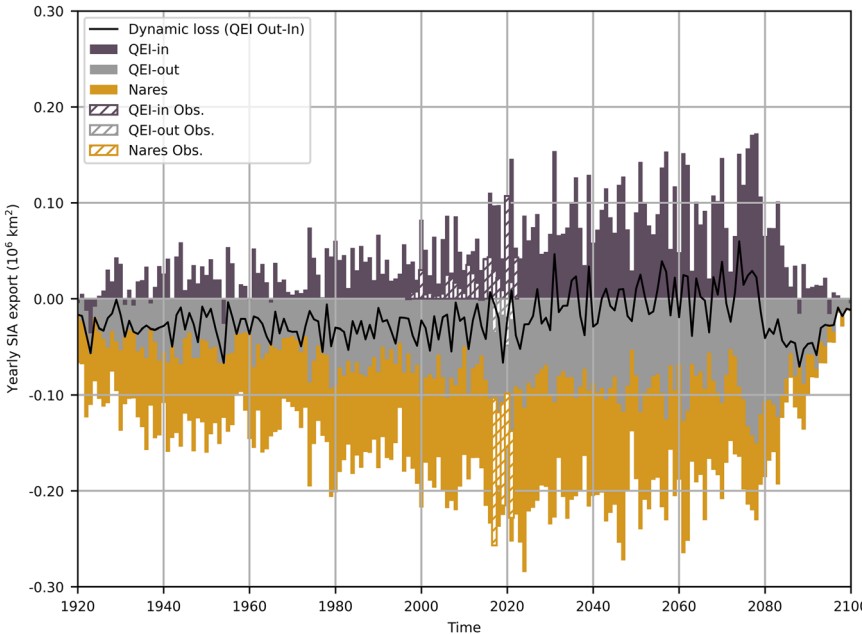

Historically, $0.15 \times 10^6$ km$^2$ of the central Arctic sea ice is transported annually into the LIA-N by the Beaufort Gyre. This increases to a maximum of ~$1.2 \times 10^6$ km$^2$ annually by the end of the century in a lower emission forcing scenario, vs. decreasing to nearly zero in high-end scenarios because the ice melts before reaching the LIA-N[9]. Across all scenarios, CMIP6 models confirm that this sea ice originating from the central Arctic Ocean will be thin and seasonal[1]. Thin central Arctic sea ice, together with reduced sea ice transport through Fram Strait (Fig. 5g), and increased melting from warmer temperatures, will not be enough to maintain a perennial ice cover in the LIA, given increased fluxes through the QEI and Nares Strait. The large increase in ice transport through the QEI suggests that sea ice in the CAA is only an effective barrier when thick, concentrated, and landfast.

Our results suggest that the last 1.2 million km$^2$ can be drained through the channels of the CAA and Nares Strait in 6–24 years. The 6 years are derived from the projected SIA fluxes from the 2040–2080 time period. The average throughflows exported through the QEI and Nares Strait for the 1920–1980 and 2040–2080 time periods are ~$(0.02 + 0.08) \times 10^6$ km$^2$ per year and ~$(0.1 + 0.11) \times 10^6$ km$^2$ per year (Fig. 6). This translates to a change of the mean residence time of sea ice in the $1.2 \times 10^6$ km$^2$ LIA-N from 12 years (=1.2/0.1 years) to 6 years (=1.2/0.21 years). This reduced residence time of sea ice through the channels of the CAA and Nares Strait could lead to the drainage of the LIA in as quickly as 6 years following continuous seasonally ice-free conditions in the central Arctic Ocean and upon ice being sufficiently thin to allow unobstructed transport (see Fig. 6). The very-conservative 24 years are derived from the observed estimates from the historical (1997–2022) period, which are about half the simulated values for the same period. Therefore, we infer that the LIA could disappear through transport in a little more than one decade after the central Arctic continuously reaches seasonally ice-free conditions. Note that this conclusion is drawn from the dynamic sea ice mass budget only and is in agreement with recent projections of sea ice extent using CMIP6 models[1].

The fate of the LIA as a whole depends chiefly on sea ice conditions in its northern part, which hinders sea ice transport and allows replenishment of thick sea ice in the QEI. In high-emission scenarios (commonly referred to as "business-as-usual", carbonbrief.org), such as the RCP8.5 used in the CESM1.3-HR, all global climate models lose sea ice of the LIA-N in the 21st century, while two thirds (22 out of 30 models) lose their multi-year ice covering under moderate emissions scenarios. All models retain significant multi-year ice in lower emissions scenarios where sea ice melting, export and ridging rates are in balance[1,2]. Whether such warming scenarios are realistic

is still a subject of debate. Schwalm et al.[38] suggest that RCP8.5 is the scenario of choice for climate change studies and impact until 2050. Hausfather and Peters[39] and the Intergovernmental Panel on Climate Change[40] consider intermediate forcings as more realistic, while RCP8.5 still being considered to be plausible.

Future work should include running similarly high-resolution climate models under lower emission scenarios to assess the sensitivity of the persistence of the LIA to shared socioeconomic pathways. Further research is also needed to identify the factors influencing sea ice trends in the current CESM1.3-HR simulation, along with insights from various high-resolution climate models, to better understand how resolution affects feedbacks and sea ice processes, and to provide a more accurate prognosis for the LIA. As one example, the number of months each summer that the LIA will remain ice-free during the transition to complete seasonal loss is deferred to future work.

The LIA has been envisioned as a refugium during the period between the current warming and ice melting and when we are able to stabilize temperatures and even trend back toward historical conditions[9]. While there are limitations to this analysis in terms of model biases, findings suggest that for the LIA to function as a bridge to the long-term survival of ice-obligate and ice-dependent species, actions are needed to reduce warming such that the central Arctic Ocean maintains some perennial thick sea ice cover.

## Methods
### CESM1.3-HR data
We use output diagnostics from the 10-ensemble members high-resolution Community Earth System Model version 1.3 (CESM1.3-HR) produced for the International Laboratory for High-Resolution Earth System Prediction (iHESP) by the Qingdao National Laboratory for Marine Science and Technology (QNLM), Texas A&M University (TAMU), and the U.S. National Center for Atmospheric Research[26,27]. The sea ice and ocean components have a nominal horizontal resolution of 0.1°(2.5–5 km in the LIA), and resolves transport through the main narrow passages of the CAA with gates wider than two grid points (Figs. 1, S8a). The Community Ice Code version 4 (CICE4) and the Parallel Ocean Program version 2 (POP2) are discretized using a B-grid on a tripolar grid, with two numerical poles located over northern Canada and Russia avoiding the singularity at the North Pole, and a regular spherical polar grid in the Southern Hemisphere. POP2 has 62 vertical levels with a 10 m resolution in the first 155 m,

gradually coarsening to 250 m near the maximum depth of 6 km[26,27,41]. The ocean model has a 3rd order upwind advection scheme, a K-profile vertical mixing parameterization[42], and explicitly resolves eddy-induced vertical transport. The sea ice component includes the elastic-viscous plastic rheology[43], a subgrid scale ice thickness distribution with 5 categories[44,45], and an energy-conserving thermodynamic scheme that accounts for the effect of internal brine pockets. The atmospheric component of the model is the Community Atmosphere Model version 5, with 30 vertical levels and a spatial resolution of 0.25°. The land component is the Community Land Model version 4 (CLM4). The simulations were initialized using a 500-year pre-industrial control simulation, followed by a 1850–2005 historical and a 2006–2100 representative concentration pathway 8.5 (RCP8.5)—i.e., a high-end - forcing scenario[26]. To this day, the monthly diagnostic outputs used for this paper are available for two of 10 ensemble members from 1920 to 2100.

## CESM1.3-LR data
As a reference, a lower resolution version of the same model (CESM1.3-LR) was run for one ensemble member on a grid with a displaced North Pole located over northern Greenland (Fig. S8b). The sea ice and ocean components of CESM1.3-LR have a nominal horizontal resolution of 1° and resolve some of the channels of the CAA. Some free parameters in the simulation were tuned to achieve an acceptable ocean thermohaline circulation, ocean, and sea ice mass balances, and top-of-atmosphere radiative balance, with the goal of consistency with CESM1.3-HR. The ocean model (POP2) has 60 vertical levels and parametrizes ocean overflow over deep channels and continental shelves[46], along-isopycnal transport from mesoscale eddies[47], and submesoscale eddies[48]. The initial conditions and forcing scenarios are the same as CESM1.3-HR. Monthly diagnostic outputs are available from 1850 to 2100. Since a formal comparison between CESM1.3-HR and CESM1.3-LR is limited due to the small number of ensemble members in both configurations, we include CESM-LR (and CESM-LE, see below) for reference only. CESM1.3-LR has a significant cold-bias in global surface temperature originating from an implicit para-metrization of vertical heat transport[26]. A detailed comparison between CESM1.3-HR and CESM1.3-LR and the effects of increased spatial resolution on output diagnostics can be found in Chang et al.[26].

## CESM2-LE data
We use the 100-ensemble members Community Earth System Model (version 2) Large Ensemble (CESM2-LE)[49]. The sea ice and ocean components have a nominal horizontal resolution of 1°, which resolve some of the channels of the CAA. The Community Ice Code version 5 (CICE5) and the POP2 are discretized on a spherical grid, with a displaced North Pole located over northern Greenland (Fig. S8c). The ocean model has 60 vertical levels with a 10 m resolution in the first 20 levels and extends to 250 m at depth. The atmospheric component of the model is the Community Atmosphere Model version 6 (CAM6), with a nominal horizontal resolution of 1° and 32 vertical levels. The land component is the Community Land Model version 5. The simulations cover the period 1850–2100 under CMIP6 historical and SSP370 future (2014–2100) forcing scenarios. Groups of ensembles were initiated using different oceanic and atmospheric initial conditions with macro- or micro-perturbations. Monthly output diagnostics are available for all 100 ensemble members.

## SAR derived sea ice area fluxes data
We use satellite-derived monthly mean SIA fluxes through entry and exit gates of the CAA and Nares Strait (Fig. 1) to quantify the amount of pan-Arctic sea ice being exported through these gateways and compare with that of CESM1.3-HR. This includes gates of the QEI, M'Clure Strait, Lancaster Sound, Nares Strait, Amundsen Gulf, and Fram Strait (Fig. 1) ranging for the 1997–2022 time period to document biases in the simulated monthly mean SIA fluxes from the CESM1.3-HR (see Table 1 for references and time periods). Monthly SIA fluxes derived from sequential pairs of synthetic aperture radars (SAR) imagery (i.e., RADARSAT-1/2, Sentinel-1AB, RADARSAT Constellation Mission) have uncertainties of ~100–500 km²

## Table 1 | SAR derived sea ice area fluxes datasets

| Gate | Time period | Reference |
|---|---|---|
| QEI-N[a] | 1997–2022 | 11,15,25 |
| QEI-S[b] | 1997–2022 | 11,15,25 |
| QEI-Out | 2016–2022 | 15 |
| M'Clure Strait | 1997–2022 | 11,25 |
| Amundsen Gulf | 2016–2022 | 15 |
| Lancaster Sound | 2016–2022 | 15 |
| Nares Strait | 2016–2021 | 12 |
| QEI-N[a] | 1997–2002 | 37 |
| QEI-S[b] | 1997–2002 | 37 |
| M'Clure Strait | 1997–2002 | 37 |
| Amundsen Gulf | 1997–2002 | 37 |
| Fram Strait | 1935[c]–2014 | 50 |

[a]QEI-N gates include Sverdrup and Parry Channel.
[b]QEI-S gates include Ballantyne, Wilkins, Pr. Gustaf Adolf.
[c]Data prior to 2004 was obtained from mean sea level pressure data retrieved from station observations across Fram Strait.

per month[15,25] and ~100 km² per month[25] through gates of the CAA (QEI-in, QEI-out, Amundsen, M'Clure, Lancaster) and Nares Strait respectively. Monthly SIA fluxes through Fram Strait are derived from SAR imagery, monthly mean sea level pressure, and buoy data that lead to uncertainties of ~850 km² per month[50].

## Sea ice concentration data: climate data records and Canadian Ice Service
We use two sets of sea ice concentration datasets for comparison of the model with observations: the National Snow and Ice Data Center (NSIDC) Climate Data Record (CDR) and the Canadian Ice Service (CIS) Digital Archive ice charts. We use the CDR sea ice concentration data to complete the CIS data in the region north of the CAA and to account for uncertainties in observational datasets.

The monthly NSIDC CDR (version 4) sea ice concentration is stored on a 25 × 25 km polar stereographic grid centered on the North Pole from 1979 to 2023[51]. The CDR product is derived from passive microwave measurements from the Nimbus-7's Scanning Multichannel Microwave Radiometer (SMMR), the Special Sensor Microwave/Imager (SSM/I) of the Defense Meteorological Satellite Program's satellites (DMSP) and the Special Sensor Microwave Imager/Sounder (SSMIS) of DMSP-F17 satellite. The sea ice concentration was generated using the Bootstrap algorithm[52] and the NASA Team algorithm[53] taking advantage of the respective strengths of each individual dataset and algorithm[7]. In this dataset, the pole hole is filled with sea ice concentration of 100%. Regional mean errors in passive microwave sea ice concentration are of 5–10% in winter and reach 30–40% in the melt season due to the presence of melt ponds, coarse-grained snow, wet snow, and refrozen surfaces. Errors are therefore greater in the marginal ice zone, where ice thermodynamic processes are key to changing sea ice conditions[54,55].

We use the gridded version of the regional CIS Digital Archive ice charts from the Eastern and Western Arctic regions, covering the QEI and the southern CAA, for comparison with CESM1.3-HR. Sea ice analysts of CIS produce ice charts for different polygons of sea ice sharing similar properties from visual interpretation of SAR imagery being the primary data source since 1995[18]. This data is discretized on a 10 × 10 km Equal Area Scalable Earth (EASE) grid from May 1982 to December 2020. Note that land masks vary between ice charts resulting in variability in sea ice extent in fully covered ice regions. Total sea ice concentration is reported using the World Meteorological Organization "egg code". CIS ice charts have monthly and weekly temporal resolution during winter and summer respectively prior to 2005; the temporal resolution is weekly all year round thereafter.

### Sea ice thickness data: PIOMAS

We use the PIOMAS assimilated sea ice concentration and derived ice thickness distribution estimates in the Arctic from 1978 to 2022[56] for comparison with the model ice thickness distribution. It uses a viscous–plastic ice rheology[24] and an ice thickness distribution with 12-ice categories[56] that conserves ice mass[24]. Known biases include thinner ice in thick ice regions, particularly along the steep thickness meridional gradient north of Greenland and of the CAA, and thicker ice in thinner ice regions[56].

### Integrated quantities

The division of the LIA into three regions centers on the QEI—the main landfast reservoir of multi-year ice—and its interaction with the LIA-N—a source reservoir to the North—and the CAA-S—a downstream reservoir to the south. Note that models and satellite-based observational products use different landmasks resulting in some differences in integrated quantities, particularly in the CAA where the ocean perimeter to area ratio is large.

### Sea ice extent, thickness, and concentration

The minimum September sea ice extent is calculated as the sum of all grid cell areas with at least 15% sea ice concentration, while the SIA is calculated as the sum of the product of sea ice concentration and area for all grid cells in a given region. The mean sea ice concentration timeseries are computed for grid cells with at least 15% sea ice concentration. The maximum mean May ice thickness is calculated from all grid cells with at least 15% sea ice concentration. The 20-years mean ice thickness distribution is calculated from weighted averages of SIA in each of the 5-ice thickness categories (0, 0.64, 1.39, 2.47, 4.57+ m) for all grid cells with at least 15% sea ice concentration. The 20-year mean ice thickness distribution for PIOMAS is calculated from the 12-ice thickness categories (-0.1, 0.1, 0.43, 1.0, 1.93, 3.30, 5.17, 7.60, 10.61, 14.18, 18.30, 22.94, 28.04+ m) for all grid cells with at least 15% sea ice concentration, then recalibrated capping the first level to 0.1 m (i.e removing the open-water fraction represented by the −0.1 m category) and summing all levels thicker than 5.17 m to ease the comparison with the model. Unrealistic values of mean sea ice thickness in CESM1.3-HR reaching 120 m in the Nansen and Eureka Sounds, i.e., well beyond the observed maximum keel depth of 25 m[17], were not filtered as they had minimal impact on the overall thickness distribution.

### Sea ice area fluxes at key gates

We calculate the dynamic contribution to SIA loss through the QEI, Nares Strait, and the CAA-S from the annual sea ice transport at key gates. Gates are identified using a digital differential analyzer (DAA) algorithm from the coordinates of its endpoints (Fig. 1). The accuracy of gate points selected by the algorithm were verified manually, and adjustments were made if needed. Most of the gates within the QEI and CAA-S are resolved in the CESM1.3-HR, except for Hell gate, and the absence of a connection between Nansen and Eureka Sounds (Table 2). The QEI-in gates include Ballantyne Strait, Wilkins Strait, Prince Gustaf Adolf Sea, Peary Channel, and Sverdrup Channel, and the QEI-out gates include Fitzwilliam Strait, Byam Martin Channel, Penny Strait, Cardigan, and Hell Gate, as defined in Fig. 1.

SIA fluxes (km² per month) at each gate are calculated using sea ice concentration, based on the method used in deriving the observed flux datasets[11,12,15,25,37,50] as follows:

$$F_i = \sum_{i=1}^{n} SIC_i U_i \Delta x,$$

where n is the number of grid edges in a gate, SIC is the sea ice concentration, $\Delta x$ (km) is the length of a grid cell segment and $U_i$ is the velocity component normal to the grid cell segment (km per month), interpolated to the center of an edge. The zonal and meridional velocities across a bipolar fold, between the two North-Hemisphere poles in the CESM1.3-HR grid near the Prince Gustaf Adolf gate, are adjusted by a factor of $1/-1$ to account for a discontinuity in the sign of the normal velocity vector. Ice fluxes are positive

**Table 2 | Gates lengths in CESM1.3-H**

| Gate | Length (km) |
| --- | --- |
| Amundsen | 174.5 |
| M'Clure Strait | 171.1 |
| Nares | 32.5 |
| Lancaster | 84.5 |
| Jones | 42.7 |
| Fram | 671.0 |
| QEI-In | 337.9 |
| Ballantyne | 52.3 |
| Wilkins | 36.5 |
| Prince Gustaf Adolf | 99.8 |
| Peary | 93.8 |
| Sverdrup | 55.5 |
| QEI-Out | 117.9 |
| Cardigan | 9.7 |
| Penny | 35.6 |
| Byam Martin Channel | 44.7 |
| Fitzwilliam | 27.9 |

when sea ice is advected in the dominant direction; either southward or into the CAA.

### Sea ice thermodynamic and dynamic tendency terms

The thermodynamic and dynamic SIA loss, from the fully ice covered, pre-melt conditions to the September SIA, are calculated from the thermodynamic and dynamic SIA tendency terms (standard CESM output diagnostics) using the melt onset and freeze onset dates. The onset dates (in Julian days) in each region are calculated as the zero-crossing of the integrated monthly mean thermodynamic SIA tendency using a simple linear interpolation (0 cm per day—SIA melt to growth or growth to melt). Total integrated melt season thermodynamic and dynamic SIA tendencies are calculated by summing the product of area tendency (% per day) and the area of the grid cells (m²) for a given region, from the beginning to the end Julian days of the melt season. The area tendency is decomposed into advection and ridging terms, the latter redistributing the thickness distribution allowing SIA to maintain or decrease under dynamical processes. Note that the SIA loss from fluxes at the gates of the QEI and the CAA-S agrees with the SIA loss from the integrated melt season advective term of the dynamic area tendency with small errors coming from the interpolation of quantities.

### Data availability

Outputs from the iHESP project[26,27] (CESM1.3-HR, CESM1.3-LR) are publicly available through the iHESP data archive (https://ihesp.github.io/archive/products/ds_archive/Sunway_Runs.html)and documented in Chang et al.[26]. The CESM2-LE data is available on the NSF National Center for Atmospheric Research platform (https://www.earthsystemgrid.org/dataset/ucar.cgd.cesm2le.output.html). The NSIDC-CDR sea ice concentration[51] and PIOMAS sea ice thickness and concentration[57] are available online. The CIS gridded ice charts are derived from weekly ice charts available on: Archive Search (ec.gc.ca). All data used for the figures is publicly available on 10.5281/zenodo.14042445.

### Code availability

Python codes of the analysis and for creating the figures are available on https://github.com/madfol97/MFOL_LIA_22_24.git.

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

## Acknowledgements
This project is a contribution to the ArcNAV project entitled: Arctic Robust Communities-Navigating Adaptation to Variability funded by the National Science Foundation—Office of Polar Program (project \#1928126), a Discovery grant from the Natural Sciences and Engineering Research Council, and the NASA ROSE grant (Grant 80NSSC20K1259) awarded to Tremblay. We acknowledge the help from Jaison Kurian at the Department of Oceanography, Texas A&M University, in accessing the CESM1.3-HR data. This work also benefited from academic and financial support of McGill University and Québec–Océan.

## Author contributions
Madeleine Fol did the data analysis, led the discussion at bi-monthly team meetings with all co-authors, and wrote the paper. Bruno Tremblay provided the day-to-day supervision for the project, participated in bi-monthly team meetings with all co-authors, and edited all iterations of the paper. Stephanie Pfirman participated in the bi-monthly team meeting with all co-authors, contributed input on existing and new data analysis presented in the paper, and conducted a thorough review of the final manuscript before submission. Robert Newton participated in the bi-monthly team meeting with all co-authors, contributed input on existing and new data analysis presented in the paper, and reviewed the final manuscript before submission. Stephen Howell participated in the bi-monthly team meeting with all co-authors, contributed input on existing and new data analysis presented in the paper, and conducted a thorough review of the final manuscript before submission. Jean-François Lemieux participated in the bi-monthly team meeting with all co-authors, contributed input on existing and new data analysis presented in the paper, and reviewed the final manuscript before submission.

## Competing interests
The authors declare no competing interests.
