## [Transparent Peer Review file · Communications Earth & Environment]

Revisiting the Last Ice Area Projections from a High-Resolution Global Earth System Model

Corresponding Author: Ms Madeleine Fol

Version 0:

Decision Letter:

Dear Ms Fol,

Your manuscript titled "Revisiting the Last Ice Area Projections from a High-Resolution Global Earth System Model" has now been seen by 2 reviewers, whose comments are appended below. You will see that they find your work of some potential interest. However, they have raised quite substantial concerns that must be addressed. In light of these comments, we cannot accept the manuscript for publication, but would be interested in considering a revised version that fully addresses these serious concerns. In particular, we expect that the revised version meets the following editorial thresholds:

1. Make a compelling argument based on analyses of other available projections for the robustness and novelty of your results, and highlight the limitations of other projections.
2. Given the biases of the high-resolution simulation in regions critical to the LIA's future extent, make a strong case (based on e.g., additional analysis) for why your conclusions are not affected by these biases.

We hope you will find the reviewers' comments useful as you decide how to proceed. Should additional work allow you to address these criticisms, we would be happy to look at a substantially revised manuscript. If you choose to take up this option, please either highlight all changes in the manuscript text file, or provide a list of the changes to the manuscript with your responses to the reviewers.

When resubmitting, please provide a point-by-point response to the reviewers' comments. Please submit your responses as a separate file, distinct from your cover letter where you can add responses to the Editors' comments that you do not want to be made available to the reviewers. Word files are preferred. We recommend that any figures, tables or graphs that are included in the response to reviewers are also included in the main article or Supplementary Information.

If the revision process takes significantly longer than three months, we will be happy to reconsider your paper at a later date, as long as nothing similar has been accepted for publication at Communications Earth & Environment or published elsewhere in the meantime.

Please use the following link to submit your revised manuscript, point-by-point response to the reviewers' comments with a list of your changes to the manuscript text (which should be in a separate document to any cover letter), a tracked-changes version of the manuscript (as a PDF file) and any completed checklist:

Link Redacted

Please do not hesitate to contact us if you have any questions or would like to discuss the required revisions further. Thank you for the opportunity to review your work.

Best regards,

Joy Merwin Monteiro, PhD
Editorial Board Member
Communications Earth & Environment
orcid.org/0000-0002-3932-3603

Alireza Bahadori, PhD
Associate Editor
Communications Earth & Environment

EDITORIAL POLICIES AND FORMAT

If you decide to resubmit your paper, please ensure that your manuscript complies with our editorial policies and complete and upload the checklist below as a Related Manuscript file type with the revised article:

Editorial Policy Policy requirements
(Download the link to your computer as a PDF.)

- Behavioural and social science
- Ecological, evolutionary & environmental sciences
- Life sciences

<https://www.nature.com/documents/nr-reporting-summary.zip>

For your information, you can find some guidance regarding format requirements summarized on the following checklist: (<https://www.nature.com/documents/commsj-phys-style-formatting-checklist-article.pdf>) and formatting guide (<https://www.nature.com/documents/commsj-phys-style-formatting-guide-accept.pdf>).

REVIEWER COMMENTS:

Reviewer #1 (Remarks to the Author):

This manuscript analyzes projections of sea ice in the Last Ice Area (LIA; North of Greenland and in the Canadian Arctic Archipelago; CAA) in a high-resolution climate model. The authors find that as the sea ice thins and becomes more mobile, there is the possibility that sea ice is flushed out of the LIA through the channels of the CAA within a decade. The main claim of the paper is thus that contrary to previous results based on low-resolution climate models (which do not explicitly resolve many of the channels in the CAA), the LIA will not persist for decades after the first ice-free summer but disappear within a decade. This has consequences for the concept of the LIA as a temporary last ice refuge for ice-dependent species until emissions are reduced. Additionally, the authors examine the thermodynamic and dynamic contributions for different parts of the LIA.

The study is easy to follow, and the figures are clear. I see the main novelty of the study in for the first time analyzing projections of sea ice in the LIA in a high-resolution climate model with explicitly resolves many of the narrow straits in the CAA, and shedding light onto the changing dynamics within the region. However, I have two main comments regarding the significance of the main claim made by the authors in this study.

First, the authors claim in the abstract that their result of the LIA disappearing a decade after the first ice-free summer under a high emissions scenario is contrary to results from low resolution models. However, to my knowledge, projections of the sea ice cover in the LIA have not been assessed rigorously across the latest generation of CMIP models. Further, Figure 5d in Jahn et al. (2024) shows the September sea-ice cover in the LIA disappearing at most one to two decades after the sea-ice cover in the central Arctic in general under a high emissions scenario. This would be quite similar to the simulations with the high resolution CESM1.3 presented in this study. The authors cite previous work that analyzed projections of the LIA sea ice cover, but as far as I can see they are all based on one model (different versions of the CCSM/CESM model). To make the claim that the quick disappearance of sea ice in the LIA under a high emissions scenario is a novel feature in high resolution climate models, the authors should ideally carry out an extensive analysis of low-resolution climate models to confirm the

longer persistence of sea ice in those models.

Second, the results are all based on one high-resolution climate model, the CESM1.3-HR. I understand that this model is currently unique in its combined high resolution in the atmosphere and ocean. However, the model also shows extensive biases (as the authors discuss) and performs worse than its low-resolution version (CESM1.3-LR) when comparing the models to observed estimates, including the mean thickness and concentration in the central Arctic north of Greenland. It is also not reassuring that the high-resolution model simulates a net sea-ice area divergence in the Queen Elizabeth Islands region, whereas there is a net convergence in the observations. Thus, I am not sure that whether the future quick disappearance of sea ice in the LIA through flushing in the high-resolution model is more, or less realistic than the projection from the low-resolution model(s). Formulated differently: Is the quick flushing of sea ice in the LIA a result of explicitly resolving the straits or is it an artifact of the thickness and concentration biases? I suggest that the authors discuss how these biases might affect the main conclusions of this study.

Reviewer #2 (Remarks to the Author):

Review of Fol et al 2024

Summary

Fol et al provide results from a high-resolution global climate model simulation with a focus on trends in sea ice in the Last Ice Area, that is the region north of Greenland and including the Canadian Arctic Archipelago where sea ice is expected to persist the longest with climate warming in the coming decades. The novelty and importance of using a high-resolution model that resolves the narrow Arctic straits to allow for a more realistic simulation of sea ice dynamics is stressed, and has been lacking in lower resolution climate models to this point. The authors share results that indicate the Last Ice Area may become ice free earlier than previously indicated in lower resolution climate models, and suggest that enhanced export of sea ice southward through the straits is better captured in the higher resolution. The paper is well written and accessible, the study is novel, the authors provide evidence to support most of their claims, and I think the work is timely and will be of interest to others in the community.

There are however some known limitations in the high-resolution model run results that reduce the strength of the conclusions, and I feel the paper would benefit from addressing these issues directly (more-so than is currently done). While I still feel the work is worthy of publication, there are some areas for improvement suggested below.

General comments

-The authors do address some of the known biases in the CESM high resolution run, however I think the paper would benefit from more explicit discussion of the known biases and limitations of the HR simulation as discussed in Chang et al 2020, and how these might impact their results and conclusions. For example, Chang et al 2020 relay that the HR run has lower overall sea ice coverage in Arctic in early 20th century, more rapid sea loss, 25% greater warming in Arctic, and include the statement "Not everything is improved by enhancing model resolution, and indeed, there are aspects of the climate in HR that are worse than in LR. For example, sea-ice extent and sea-ice concentration in both hemispheres in HR are underestimated compared to LR and are less realistic." In light of these known issues in the HR run, which make Fol et al's arguments somewhat less convincing, I think the authors need to explicitly address these limitations and provide more justification for why their results are robust - or provide caveats to their conclusions accordingly. In addition, the authors should note that the differences between HR and LR extend to more than just horizontal resolution, but also include differences in physics parameterizations, parameterization settings, vertical resolution, etc, so discussion of how these aspects may affect results is warranted.

-Related to the above comment and the known issues with the HR run... the HR run has lower sea ice area (SIA) and thickness compared to the low-resolution (LR) run, and there is therefore less ice in the 'ice shed' sources to supply the ice to the LIA (i.e. along the marginal shelf seas including the Siberian shelf). As a result one could expect less ridging and thinner ice in the LIA, which will result in more mobile ice, and thus more export through the straits. Further discussion of how less and thinner sea ice in the pan-Arctic could impact results in the LIA would be beneficial.

-I think the paper would benefit from clearly providing the author's definition of the Last Ice Area (LIA) quite early in the manuscript because their definition differs somewhat from others used in previous literature, so this could be a source of confusion for readers (specifically that the authors definition includes the CAA, which is not included in some other studies)

-I feel the paper would benefit from providing more results for sea ice trends across the pan-Arctic in the HR run for context for the LIA results. Specifically, one of the main conclusions is that the LIA will disappear one decade after an ice-free pan-Arctic is reached, but the figures in the paper do not make it straight-forward to evaluate this conclusion. The authors could consider moving some of the pan-Arctic data presented in Fig. S5 into the main paper and into some of the related figures (discussed below) to better support this conclusion and provide direct evidence to allow readers to assess this conclusion.

-Is there opportunity to make more comparisons with other high-resolution sea ice / climate model runs? I personally am not familiar with the full suite of existing high resolution GCMs, but are there not some other high-resolution model runs to compare to (even regional models) to better evaluate the sea ice trends? Perhaps GFDL CM2.6 and HighResMIP CMIP6? If the authors can compare their results with other high resolution model runs this would go a long way toward strengthening their conclusions. And if not, then perhaps more emphasis in the conclusions on what further analysis of this HR run is

needed, and what other HR model runs are needed to better evaluate this results (beyond just low-emission scenario runs) would be beneficial

Specific comments:

13-14: here as phrased, the LIA seems to be the area north of Greenland AND north of CAA. This should be revised to make it explicit the authors are referring to LIA as including the whole CAA

20-21 - the statement 'the LIA could disappear a little more than one decade after Arctic Ocean has reached seasonally ice-free conditions' is presented as one of the main conclusions, but I found this conclusion difficult to assess with the figures present. See below for suggestions

30 - Tuvaijuittuq should be delineated on Fig 1

34-35 - "These Arctic wildlife are a cornerstone of Inuit hunting practices, a heritage now imperilled by the decline of sea ice." - true, but it depends on how the authors are defining the LIA and whether it includes the QEI and CAA-S. For the LIA-N alone I am not aware of any Inuit that regularly hunt in that region, so it is not clear how the loss of the ice in the LIA-N will imperil Inuit heritage (one can certainly make indirect connections here). Perhaps this statement could be generalized to discuss pan-Arctic sea ice loss and environmental changes in general, or alternatively focused to the specific impacts of sea ice loss in the LIA (ecological changes, travel/shipping hazards, and impacts to Inuit)

36 - I would suggest a clear statement of how the authors are defining the LIA - specifically that it includes the CAA which is in contrast to a number of other definitions of the LIA in the literature - is needed here early in the introduction to avoid confusion and make it clear why the authors are including the QEI and CAA in their discussion. The specific definition of the 3 regions (LIA-N, QEI, and CAA-S) that comes at the end of the introduction can still be included where it is (Lines 75-78), but a more explicit definition of the LIA as authors define it is needed earlier (including mention of what marine conservation areas are included (e.g. includes Tuvaijuittuq, what about Pikialasorsuaq/Sarvarjuaq? Tallurtiuq Imanga?...and each of these should be included in Fig 1 if mentioned in text)

85-92 - these sentences are not results, they are more a summary of methodological choices and I think would be better moved to the last paragraph of the introduction.

The results section would benefit from beginning with a brief description of pan-Arctic sea ice trends in the HR simulation, then transition into specifics about the LIA. This would help provide context for the LIA results, and also better support one of the main conclusions the authors are making related to LIA becoming ice free within one decade after ice-free Arctic.

Related to this, I would also suggest that the authors could include more discussion of the known biases and limitations of the CESM-HR simulation as it relates to Arctic sea ice and temperatures in the introduction. Provide enough information up-front so the reader can evaluate the results in this context on the first read through, and it would allow the authors to pre-emptively justify the HR results and some of their limitations. I think this information could be placed at the end of the introduction.

94-96 - how much of the sea ice loss being dominated by thermodynamic term later in the simulations could be explained by the fact that the CESM-HR is know to run hot in the Arctic? It appears there is more than just higher spatial resolution at play here that could be affecting the results.

96-98 - sentence bit confusing, maybe re-word

98-102 - again, bit hard to follow. Suggest separating concepts out into separate sentences. Focus on LIA-N for first sentence. Focus on QEI for second sentence

178-180 - Reads "This, coupled with enhanced melting, results in the complete drainage of the LIA and a seasonally ice-free Arctic, until sea ice availability in the source regions (i.e the LIA-N and central Arctic) declines (2081-2100)" but other than Fig. S5, there is no citation and little evidence given to support this statement with respect to a seasonally ice-free Arctic or decline of sea-ice in source regions. I feel perhaps the pan-Arctic data in Fig. S5 could be moved into the main paper and included in Fig. 2, 3, & 6 as separate panel. Note this might also necessitate revising Fig 1 to include a pan-Arctic map and labels (perhaps as a small inset)

201-202 - "The fate of the LIA depends chiefly on sea ice conditions north of the CAA, which is the region's ice source." - this statement is somewhat confusing as the region north of the CAA is within the LIA as per Fig 1. And the introduction says the source is actually the marginal shelf seas - please correct this apparent contradiction

210-212 - Although running HR simulations under lower emission scenarios would indeed be valuable, I would suggest that further research into what factors are affecting sea ice trends in the existing HR simulation would also be beneficial and informative

215-218 - I feel this statement "This analysis suggests that rapid reductions of emissions of greenhouse gases and carbon capture would be needed for the LIA to function as a bridge to the long-term survival of ice-obligate and ice-dependent species." is not well supported by the results presented in this paper. While the authors make inferences about what lower emission scenarios would likely show, those results are not yet available to make this conclusion here. I am also not

convinced, given the known biases in the HR model, that all the sea ice loss trends presented in this paper are robust enough to neglect the possibility that some of what has contributed to rapid sea ice loss from the LIA in the HR version could be due to these inherent biases (as opposed to simply being able to resolve the narrow straits of CAA and Nares). I suggest the authors temper their conclusion with these limitations in mind (this is still a valuable paper, and should rightfully stimulate more research into the behaviour of sea ice in the HR models and the need to improve them).

369- Please explicitly explain how sea ice 'ridging' term is defined and how it contributes to sea ice loss in the dynamic component

Fig. 1 - Three regions are defined (LIA-N, QEI, and CAA-S). My understanding is that the authors intend for all of these three regions combined to form the overarching Last Ice Area. This seems a somewhat different definition than previous literature (e.g. Newton et al, 2021), where LIA-N is closer to the standard definition, where there is overlap with Tuvaijuittuq MPA, and excludes the CAA owing to coarse resolution of LR climate models and satellite imagery that cannot resolve narrow passages. Some clarification here is required. I would suggest some over-arching outline or shading could be used to delineate the entire LIA (and labelled as such), with the 3 sub-regions also defined as at present.

It is also not clear why certain waterways, including the Nansen and Greely Sounds system, Jones Sound, and coastal waters along N Ellesmere and Greenland and the Lincoln Sea are not included in the shaded areas.

Is it worth including the outline of the 'Remnant Arctic Multi-Year Sea Ice and the Northeast Water Polynya Ecoregion' as a site for potential Outstanding Universal Value for inclusion on the UNESCO World Heritage List = Remnant Arctic Multi-Year Sea Ice and the Northeast Water Polynya Ecoregion (<https://portals.iucn.org/library/sites/library/files/documents/2017-006.pdf>)? This region seems to coincide with the LIA as defined by the authors in the present study, so could be useful to include.

Minor notes:

-A higher resolution image would improve the map quality

-“Eur.” is shown on map but not defined in caption.

-The “H” for Hells Gate should be moved a bit to the left closer to the gateway

Fig. 4 - bit confusing the time periods of the model simulations vs observation

Also the legend has time periods label (2081, 2100) - does this mean 2081 to 2100? If so perhaps label as 2081-2100?

Plotted are 20-year averages, but the 2001-2020 model results are not 20-year averages, they are broken into smaller periods to match obs. This comparison is great, but it adds some confusion. Perhaps include a full 2001-2020 average to be consistent with other 20-year periods, and then add in an additional line that only plots the shorter observational comparison period? Though I expect this would not add any meaningful difference to the plot, and perhaps just clutter it unnecessarily? I leave it to the authors to determine.

Alternatively, perhaps just state that “shown are 20-year averages, except for the 1921-1980 long-term average, and the 2001-2020 period where a subset of years are averaged to coincide with the observational records as follows...”

Fig. S2 - please indicate what the solid line and the shaded areas mean? The mean of 2 ensemble members and variance?

-there is overlap in y-axis labels

-the thermodynamic loss is related increasing air and sea surface temperatures - but the HR simulation is known to be biased warm in this regard. Further acknowledgement and discussion of this bias and its effect on the results is warranted.

Fig. S5- CESM-LR does better job with SIA, SIE, only HR does better with thickness compared to observations. This reduces how convincing the sea ice results from the HR run are, but as long as this is clearly addressed in the paper that is sufficient.

Fig. S6 - HR vastly underestimates concentration, and somewhat underestimates thickness. Again, this reduces how convincing the sea ice results from the HR run are, but as long as this is clearly addressed in the text of the paper that is sufficient. A higher resolution figure would help the image quality of this figure.

Communications Earth & Environment is committed to improving transparency in authorship. As part of our efforts in this direction, we are now requesting that all authors identified as 'corresponding author' create and link their Open Researcher and Contributor Identifier (ORCID) with their account on the Manuscript Tracking System prior to acceptance. ORCID helps the scientific community achieve unambiguous attribution of all scholarly contributions. You can create and link your ORCID from the home page of the Manuscript Tracking System by clicking on 'Modify my Springer Nature account' and following the

instructions in the link below. Please also inform all co-authors that they can add their ORCID to their accounts and that they must do so prior to acceptance.

Version 1:

Decision Letter:

Dear Ms Fol,

Your revised manuscript titled "Revisiting the Last Ice Area Projections from a High-Resolution Global Earth System Model" has now been seen by our reviewers, whose comments appear below. In light of their advice we are delighted to say that we are happy, in principle, to publish a suitably revised version in Communications Earth & Environment, provided you better define what constitutes "sea ice disappearance" in the Last Ice Area and adjust your conclusions accordingly, while also using the term "ice-free central Arctic" to avoid confusion, along the lines recommended by our referees.

We therefore invite you to revise your paper one last time to address the remaining concerns of our reviewers. At the same time we ask that you edit your manuscript to comply with our format requirements and to maximise the accessibility and therefore the impact of your work.

EDITORIAL REQUESTS:

******Please take care to match our formatting and policy requirements. We will check revised manuscript and return manuscripts that do not comply. Such requests will lead to delays. ******

SUBMISSION INFORMATION:

OPEN ACCESS:

Communications Earth & Environment is a fully open access journal. Articles are made freely accessible on publication. For further information about article processing charges, open access funding, and advice and support from Nature Research, please visit <https://www.nature.com/commsenv/open-access>

Link Redacted

Best regards,

Alireza Bahadori, PhD

Associate Editor
Communications Earth & Environment

Joy Merwin Monteiro, PhD
Editorial Board Member
Communications Earth & Environment
orcid.org/0000-0002-3932-3603

REVIEWERS' COMMENTS:

Reviewer #1 (Remarks to the Author):

The authors have done a good job in responding to my concerns by clarifying the main conclusions and by further discussing the potential impact of model biases in CESM-HR on the results. I now recommend publication. This study will be a nice addition to the literature, as it improves our understanding of the changing sea-ice dynamics in the LIA and their impact on the fate of the LIA sea ice, which is of interest for a broad field of scientific research.

Here are some very few minor comments:

L139-140: similar to that of CESM2-LE?

L214 – 215: I think the increase in mean sea-ice concentration is a bit of an artifact resulting from only selecting grid cells with >15% SIC. As seen in Fig S1, there is essentially no sea ice left in Sept. 2041-60, except for some coastal ice, which is then used to calculate the mean.

L267: As I stated previously, studies 3, 8 and 9 are all based on CCSM/CESM, so I would suggest specifying that here

Reviewer #2 (Remarks to the Author):

Fol et al. Second Review

General Comments

The authors have addressed most of the reviewer comments thoughtfully, and I thank them for the effort in considering both Reviewer #1 and my own comments. However, I am still left with a few questions:

1) The main conclusion of the study is that the LIA will disappear within one decade after an ice-free Arctic is reached. The metric for an ice free Arctic ($<1,000,000\text{km}^2$) is presented, but what is the metric for 'disappearance' of sea ice in the LIA?

I can only assume the authors mean zero sea ice extent in the LIA? And related to this, although the inclusion of the pan-Arctic data in Fig 2 is an improvement, I still do not think it clearly shows that the LIA will disappear within 1 decade after ice-free Arctic. I am left pondering whether this conclusion is well-supported by the data on-hand. Fig. 4 shows the pan-Arctic dropping to ice-free in 2035, and maybe the LIA-N dropping to zero sometime between 2040-2050, but QEI not until after 2060 (and CAA-S is not part of LIA). So unless the metric is something higher than zero SIA, I am left confused. Perhaps demarcating (vertical or horizontal lines) on these figures when these milestones are reached is needed to clarify the timing? Quite frankly, in my interpretation of Fig. 2, I do not see that the LIA-N, or QEI, or CAA-S reach zero extent within one decade after pan-Arctic drops to ice free conditions. So I feel the authors really need to clearly, and unambiguously, lead the reader to the evidence for this conclusion.

On a related note, on the metric of $<1,000,000\text{km}^2$ as ice-free Arctic. Presumably the remaining sea-ice that makes up that last $1,000,000\text{km}^2$ is that which is found in the LIA. So in fact the authors are concluding that the LIA will disappear within one decade after the LIA is the only sea ice left in the Arctic. There is some circular reasoning here that is perhaps unnecessary, but maybe an unfortunate artifact of the accepted definition of an ice-free Arctic. It might be prudent to consistently use the term 'ice-free central Arctic' to distinguish the regions being referred to, and help reduce the circularity.

2) The authors have done well to include more discussion of the biases of the model, and have presented more background literature in the introduction. However, in the introduction, when referring to Jahn et al. 2024, the authors state: "This thick sea ice disappears 10 years to 20 years after reaching a consistent seasonally ice-free central Arctic under moderate to high warming scenarios." To me this seems to indicate that the main conclusion has already been presented in the literature, which appears to reduce the novelty of the current manuscript. The authors very clearly need to state how their work is novel and distinct (in methods or insight) if the basic conclusion is not much different than previous literature.

3) There remain some confusing structural or grammatical issues in the text. I recommend the authors review the paper to ensure all text is clear and unambiguous, especially the transitions where new sentences have been inserted. There are some minor edits that would help, I have suggested a few below, but please review the full paper for clarity, readability, and flow.

Specific comments

25 - I feel these opening statements need some minor clarification to emphasize the importance and characteristics of the LIA - particularly to highlight that the LIA area is where the oldest, thickest sea ice is, and is expected to persist for some amount of time even after the rest of the Arctic becomes seasonally ice free. Perhaps consider revising this opening sentences to something like: "Within a few decades, most of the Arctic Ocean is expected to become seasonally sea ice-free. The exception to this trend is the region surrounding the QEI and areas north of the CAA and Greenland, where the oldest, thickest sea ice accumulates, and is where perennial sea ice cover is expected to persist (for some time period) even after the remainder of the Arctic is ice free in summer. This region has been called the Last Ice Area (LIA), a concept that was first..."

Some revision along these lines would be beneficial, but please rewrite as the authors see fit.

27 - 'This region was first.' = the region was already there, it is more the concept of the LIA that was introduced at AGU (which is how it was stated in the original submission) - but the phrase 'Last Ice Area' needs to be presented prior to stating that the concept was introduced, as above.

38 - 'This analysis provides...' - some uncertainty which analysis the authors are referring to, presumably the present study, but this sentence would benefit from some clarification.

50-52 - these two new sentences are somewhat out of place as the opener to a paragraph that is mainly about sea ice circulation. Would suggest moving these sentence into paragraph starting at Line 84, perhaps somewhere around Line 86.

52 - 'and the northern Baffin Bay.' - remove 'the'

55-59 - four advective pathways are listed, but there is no mention of possibility of thermodynamic loss in the LIA. If this paragraph is focus on dynamic processes then please state as much in the paragraph opening, otherwise consideration of thermodynamic process is warranted.

And regardless of whether it is presented here or elsewhere, I feel that further discussion of factors influencing thermodynamic sea ice loss is needed, including splitting out atmospheric versus oceanic components. How much thermodynamic loss is due to surface melting versus basal melting? Can this tell us anything about whether it is more important for the climate models to get the atmospheric warming right, or the sea surface temperature/salinity right?

65-69 - This paragraph seems out of place and incomplete unto itself. Perhaps it can be combined with the paragraph below (70-83)?

80 - with increasing or decreasing sea ice concentrations? (presumably increasing concentrations, but please explicitly state as much)

95-96 - this sentence need revising...present and interpret what exactly about the LIA? in context of what biases?

106-107 - please explicitly state that these are known bias in the literature, and include an appropriate citation

112 - I feel this paragraph needs a concluding sentence to address how these biases might impact the results and conclusions that will follow, or some statement saying that these biases will be discussed further later in the paper.

122 - 'around 2025 and 2067.' - do the authors mean 'between 2035 and 2067'?

123-125 - this sentence seems to state that the main conclusion of this paper actually already exists in the literature

240 - until 2000 = is this the correct year? This seems confusing because the previous sentence refers to 2040-2060 time period, and this sentence begins with 'This gradual loss..' indicating that the authors are referring to the 20240-2060 time period.

246 - 'complete drainage of the LIA.' By when? Add a date (or duration after ice-free Arctic if preferred)

265 - drainage of the LIA in as quickly as 6 years.' - where are the author getting this interpretation from? Can they refer to a specific figure? I presume it is from Line 250, the 6 year residence time, but if so please be explicit in this connection as this is 6 years is presented in a different paragraph, so the connection is tenuous. My preference is that the authors can refer to a specific figure (Such as Fig 2) showing ice in the LIA disappearing with 6 years after ice-free pan-Arctic, but right now I fail to see this clearly in any figure.

267-273 - the authors estimate disappearance of the LIA in a little over a decade after ice-free Arctic, and state (2 x 6 years) - but where are they getting the 2 x 6 years. I think I see where the 6 years is from, but why multiplying by 2? What is the justification?

271 - 'or that of the future.' = this seems confusing, how do you double sea ice fluxes of the future?

274 - 'and given that' - This sentence needs revising.

Fig. 1 - the revisions to Fig 1 are appreciated in how they present the various regions and flux gates. However I think the figure could be improved in two ways:

- 1) consider using a different projection, or at least a rectangular aspect ratio to the map, in order to make better use of space (right now over half the map is 'wasted' space (i.e. the lower right-hand ~1/3 of the map is not utilized), and to centre and focus the map on the LIA/QEI/CAA
- 2) consider moving the flux gate labels off to the side and draw thin lines connecting the label to the strait of interest, thus reducing the clutter of labels, the need for text background shading and text box outlines, and the fact that they obscure geographic features

Fig. 2 & 4 - I feel some demarcation showing clearly the timing of the disappearance of sea ice in the LIA is needed, to allow the reader to see that this occurs within 1 decade after ice-free central Arctic.

Response to reviewers - Revisiting the Last Ice Area Projections from a High-Resolution Global Earth System Model

Madeleine Fol, Bruno Tremblay, Stephanie Pfirman, Robert Newton, Stephen Howell, Jean-François Lemieux

Dear reviewers,

We want to thank you for a careful review of the manuscript, which has led to a much improved and clearer manuscript. You will find below the point-by-point responses to each of the reviewer's comments in blue and black respectively. We hope that the proposed changes will respond to the reviewers' concerns.

Point-by-point revisions from reviewers' comments:

Reviewer #1 (Remarks to the Author):

We thank Reviewer #1 for their careful review. Addressing the comments led to a much-improved revised manuscript. Specifically, the goal of the study and the uncertainty in the interpretation of the results in the light of known bias in the model is now more clearly stated.

This manuscript analyzes projections of sea ice in the Last Ice Area (LIA; North of Greenland and in the Canadian Arctic Archipelago; CAA) in a high-resolution climate model. The authors find that as the sea ice thins and becomes more mobile, there is the possibility that sea ice is flushed out of the LIA through the channels of the CAA within a decade. The main claim of the paper is thus that contrary to previous results based on low-resolution climate models (which do not explicitly resolve many of the channels in the CAA), the LIA will not persist for decades after the first ice-free summer but disappear within a decade. This has consequences for the concept of the LIA as a temporary last ice refuge for ice-dependent species until emissions are reduced. Additionally, the authors examine the thermodynamic and dynamic contributions for different parts of the LIA.

The study is easy to follow, and the figures are clear. I see the main novelty of the study in for the first time analyzing projections of sea ice in the LIA in a high-resolution climate model with explicitly resolves many of the narrow straits in the CAA, and shedding light onto the changing dynamics within the region. However, I have two main comments regarding the significance of the main claim made by the authors in this study.

First, the authors claim in the abstract that their result of the LIA disappearing a decade after the first ice-free summer under a high emissions scenario is contrary to results from low resolution models. However, to my knowledge, projections of the sea ice cover in the LIA have not been assessed rigorously across the latest generation of CMIP models. Further, Figure 5d in Jahn et al. (2024) shows the September sea-ice cover in the LIA disappearing at most one to two decades after the sea-ice cover in the central Arctic in general under a high emissions

scenario. This would be quite similar to the simulations with the high resolution CESM1.3 presented in this study. The authors cite previous work that analyzed projections of the LIA sea ice cover, but as far as I can see they are all based on one model (different versions of the CCSM/CESM model). To make the claim that the quick disappearance of sea ice in the LIA under a high emissions scenario is a novel feature in high resolution climate models, the authors should ideally carry out an extensive analysis of low-resolution climate models to confirm the longer persistence of sea ice in those models.

#1. The reviewer is correct, a rigorous assessment of LIA projections from low resolution climate models has not been published yet. The Introduction now gives a more detailed review of previous projections and uncertainties with respect to the timing of an “ice-free” Arctic – i.e. defined as the time when the multi-model mean sea ice extent reaches one million km² and variability around the mean – and disappearance of the Last Ice Area from CMIP3, CMIP5 and CMIP6 (Stroeve et al. 2007; 2012; Notz 2020; Jahn et al. 2024).

The key limitation of previous work is that the models do not resolve the narrow passages of the Canadian Arctic Archipelago, a “dynamical” sink term that we argue is equally important in setting the timescale for the existence of the LIA as the thermodynamic (melt) sink term.

Quantifying the impact of this dynamical sink is challenging even in the context of the High-Resolution CESM1.3 project where the same model version was run at high and low resolution because (i) inherent differences must be present in both version of the model, such as the presence/absence of parameterizations of ocean fluxes associated with small scale processes that are only resolved in HR, (ii) and equally importantly, because of feedback with the land-ocean-ice-atmosphere system that leads to a significantly different mean climate in both version of the model (see for instance Fig 2 of the revised manuscript). It is for this reason that we do not present a comparison of the absolute timing of the disappearance of the LIA with previous studies, but rather an estimate of the timescale required to drain and melt (rather than melt only) the LIA, in the CEM1.3-HR. We justify the approach and estimate the uncertainty by comparing the simulated ice area fluxes from the CESM1.3-HR with those from the satellite historical record.

The revised Introduction now reads: “These characteristics of sea ice circulation are only partly resolved in low resolution models (1 and 4). This paper examines the additional impacts of sea ice transport through the QEI and Nares Strait (2 and 3) on projections of the LIA, while also addressing larger-scale processes. ... First, we evaluate the pan-Arctic ice that feeds the LIA and assess regional biases in comparison to lower-resolution global climate models—specifically CESM1.3-LR and CESM2-LE—, the Pan-Arctic Ice Ocean Modeling and Assimilation System (PIOMAS), and observed sea ice extent, area, thickness, and fluxes derived from high-resolution satellite imagery. Then, we present and interpret the LIA in the context of these biases. Finally, we discuss the dynamic and thermodynamic contributions - and potential feedback - responsible for the disappearance of perennial sea ice of the LIA. ...

Previous studies using climate models have dedicated considerable effort into estimating a realistic timing of a seasonally ice-free Arctic, by trying to reduce major uncertainties rising from the model, climate variability and the uncertainty in future greenhouse gas scenarios. Most CMIP3 and CMIP5 models failed to reach ice-free conditions at the end of the century, decoupling from the rapid observed decline of pan-Arctic sea ice^{31,32}. The CMIP6 generation of models showed improvements in reproducing observed sea ice conditions^{1,2}. CMIP6 models predict that the central Arctic could experience its first ice-free summer as early as the 2020s or 2030s, with a high likelihood of this occurring by 2050. Under all warming scenarios, the central Arctic Ocean is expected to be ice-free in most summers, including natural variability, by mid-century, around 2035 and 2067¹. Only under the lowest warming scenario (SSP1-1.9) do CMIP6 models keep significant thick September sea ice north of the CAA at the end of the century¹. This thick sea ice disappears 10 years to 20 years after reaching a consistent seasonally ice-free central Arctic under moderate to high warming scenarios¹. As part of the CMIP6 suite, HighResMIP models that resolve the channels of the CAA and Nares Strait generally overestimate SIA loss in the LIA compared to observations, projecting both regional and pan-Arctic seasonally ice-free conditions by around 2050³³. Uncertainties regarding projections for regional ice-free conditions using CMIP models are greater than those for pan-Arctic projections and are highly sensitive to the specific models employed³⁴. Rather than attempting to project the exact timing of the regional disappearance of the LIA, we provide the fastest rate that it could take to drain and melt the LIA once the central Arctic Ocean becomes seasonally ice-free, thereby accounting for the biases and uncertainties in CESM1.3-HR.”

The main finding of the manuscript - that the LIA could disappear a little more than one decade after the Arctic Ocean has reached seasonally ice-free conditions - is only possible to draw when considering the amount of sea ice being fed into the LIA. Those were taken from lower resolution CMIP6 simulations to strengthen the estimate of the timescale required to drain and melt the LIA. The revised Discussion now reads: “Historically, $150 \times 10^3 \text{ km}^2$ of the central Arctic sea ice is transported annually into the LIA-N by the Beaufort Gyre increasing to a maximum of $\sim 1,200 \times 10^3 \text{ km}^2$ annually by the end of the century in a lower emission forcing scenario and nearly zero in high-end scenarios as it melts before reaching the LIA-N⁹. Across all scenarios, CMIP6 models confirm that this sea ice will be thin and seasonal¹. Reduced sea ice transport through Fram Strait (Fig. 5g), and increased melting from warmer temperatures, indicate that thin pan-Arctic sea ice will not be enough to maintain a perennial ice cover in the LIA, given the fluxes through the QEI and Nares Strait. Consequently, this enhanced ice transport through the channels of the CAA and Nares Strait could lead to the drainage of the LIA in as quickly as 6 years following continuous seasonally ice-free conditions in the central Arctic Ocean, and upon ice being sufficiently thin to allow unobstructed transport. ”

The offensive part of the sentence in the abstract, “...contrary to results from various lower resolution models” was removed. Abstract now reads: “Under this potentially worst-case scenario, the LIA could disappear a little more than one decade after the central Arctic Ocean has reached seasonally ice-free conditions.”. The disappearance of the LIA is not a novelty rising from a high-resolution model; but the mechanism behind the timing of its disappearance is.

Second, the results are all based on one high-resolution climate model, the CESM1.3-HR. I understand that this model is currently unique in its combined high resolution in the atmosphere and ocean. However, the model also shows extensive biases (as the authors discuss) and performs worse than its low-resolution version (CESM1.3-LR) when comparing the models to observed estimates, including the mean thickness and concentration in the central Arctic north of Greenland. It is also not reassuring that the high-resolution model simulates a net sea-ice area divergence in the Queen Elizabeth Islands region, whereas there is a net convergence in the observations. Thus, I am not sure that whether the future quick disappearance of sea ice in the LIA through flushing in the high-resolution model is more, or less realistic than the projection from the low-resolution model(s). Formulated differently: Is the quick flushing of sea ice in the LIA a result of explicitly resolving the straits or is it an artifact of the thickness and concentration biases? I suggest that the authors discuss how these biases might affect the main conclusions of this study.

#2. While there is a bias low in pan-Arctic SIA in CESM1.3-HR, the sea ice concentration in the LIA-N in both high- and low-resolution models are similar because of the omni-present sea ice convergence against the CAA coastline – see Figure just below, where the mean sea ice concentration time series in the LIA-N are shown for CESM1.3-HR and CESM1.3-LR. Since sea ice resistance is mostly dependent on sea ice concentration (Hibler, 1979), thicker sea ice would lead to similar sea ice area fluxes in the LIA. This was clarified in the revised manuscript with the mention “results not shown”.

Mean september sea ice concentration for grid cells with at least 15% sea ice concentration (dashed) and for all grid cells (full lines) for CESM1.3-HR (dark blue) and CESM1.3-LR (light blue) in a) the LIA-N, b) the QEI and c) the CAA-S

The revised result section 2.1 now reads: “While there are negative biases in sea ice extent, SIA and sea ice thickness in CESM1.3-HR, the sea ice concentration in the LIA-N is similar to that of CESM1.3-LR (not shown), because of the omni-present sea ice convergence against the northern CAA coastline. Since sea ice resistance is mostly dependent on ice concentration²⁵, thicker sea ice would lead to similar sea ice area fluxes in the LIA.”

The observed annual SIA convergence of $9 \times 10^3 \text{ km}^2$ for the 2017-2021 period is within the envelope of natural variability (80e percentile) simulated by CESM1.3-HR (ranging from -

97x10³ km² to 23x10³ km² with the simulated annual SIA divergence of 21x10³ km²) evaluated for the same 2017-2021 5-years period. Also, observations have important uncertainty, as stated in the online methods section: “Monthly SIA fluxes derived from sequential pairs of SAR imagery (i.e. RADARSAT-1/2, Sentinel-1AB, RADARSAT Constellation Mission) have uncertainties of ~100-500 km²/month^{14,22} and ~100 km²/month²² through gates of the CAA (QEI-in, QEI-out, Amundsen, M’Clure, Lancaster) and Nares Strait respectively.”

Reviewer #2 (Remarks to the Author):

We thank Reviewer #2 for their positive review and thorough comments on the manuscript. Addressing the comments led to a much improved revised manuscript. Specifically, we clarified the definition of the LIA, gave a detailed review of previous projections and uncertainties in the timing of an “ice-free” Arctic and disappearance of the Last Ice Area from CMIP3, CMIP5 and CMIP6 (Stroeve et al. 2007;2012; Notz 2020; Jahn et al. 2024), and included a sea ice concentration analysis allowing a clear assessment of the sea ice fluxes through the QEI (see Fig S2 of the revised manuscript), a limit that is reached when sea ice interactions are small (i.e. sea ice concentration is below 90%) and whose magnitude depends chiefly on the simulated wind forcing by the model.

Summary

Fol et al provide results from a high-resolution global climate model simulation with a focus on trends in sea ice in the Last Ice Area, that is the region north of Greenland and including the Canadian Arctic Archipelago where sea ice is expected to persist the longest with climate warming in the coming decades. The novelty and importance of using a high-resolution model that resolves the narrow Arctic straits to allow for a more realistic simulation of sea ice dynamics is stressed, and has been lacking in lower resolution climate models to this point. The authors share results that indicate the Last Ice Area may become ice free earlier than previously indicated in lower resolution climate models, and suggest that enhanced export of sea ice southward through the straits is better captured in the higher resolution. The paper is well written and accessible, the study is novel, the authors provide evidence to support most of their claims, and I think the work is timely and will be of interest to others in the community.

There are however some known limitations in the high-resolution model run results that reduce the strength of the conclusions, and I feel the paper would benefit from addressing these issues directly (more-so than is currently done). While I still feel the work is worthy of publication, there are some areas for improvement suggested below.

General comments

-The authors do address some of the known biases in the CESM high resolution run, however I think the paper would benefit from more explicit discussion of the known biases and limitations of the HR simulation as discussed in Chang et al 2020, and how these might impact their results and conclusions. For example, Chang et al 2020 relay that the HR run has lower overall sea ice coverage in Arctic in early 20th century, more rapid sea loss, 25% greater warming in Arctic, and include the statement “Not everything is improved by enhancing model

resolution, and indeed, there are aspects of the climate in HR that are worse than in LR. For example, sea-ice extent and sea-ice concentration in both hemispheres in HR are underestimated compared to LR and are less realistic.“ In light of these known issues in the HR run, which make Fol et al’s arguments somewhat less convincing, I think the authors need to explicitly address these limitations and provide more justification for why their results are robust - or provide caveats to their conclusions accordingly. In addition, the authors should note that the differences between HR and LR extend to more than just horizontal resolution, but also include differences in physics parameterizations, parameterization settings, vertical resolution, etc, so discussion of how these aspects may affect results is warranted.

#3. Different models have different biases in sea ice extent. Smaller biases, however, do not imply a more realistic mean climate or sensitivity to warming. For instance, the CESM1.3-LR has much lower biases in SIE compared with CESM1.3-HR, but this is accomplished by a cold bias in global surface temperature originating from an implicit parametrization of vertical heat transport, which reduces the sensitivity to Arctic warming. It is for this reason that we report on the time scale for the disappearance of the LIA in CESM1.3-HR starting from the moment when the Arctic is seasonally mostly ice-free with an end-of-summer relatively compact sea ice cover only present within and to the north of the CAA. This estimate relies primarily on simulated sea ice area fluxes within the CAA that compare with those of the satellite record.

To address whether the changes in magnitude of those fluxes in a warmer climate are correctly simulated, we look at the temporal evolution of the Aug-Sep-Oct sea ice concentration (SIC) in the QEI (Figure below, Fig S2 in the revised manuscript). To the first order the sea ice area fluxes through the QEI are a function of the ice-ice interaction term (i.e., the rheology term), which has an exponential dependence on SIC (a factor of 10 decrease for a change in SIC of only 10%; Hibler, 1979). The simulated Aug-Sep-Oct SIC in the QEI has a negative bias of approximately 10% when compared with that of the historical record, consistent with the bias in the observed-simulated sea ice area flux. In the second half of the record (2000-2023), the mean observed Aug-Sep-Oct SIC in the QEI is 90% (essentially in free-drift) with occasional short excursions at 96% when the ice interaction term will play a role in the momentum balance. In the 2020-2080 time period, the simulated mean Aug-Sep-Oct SIC in the QEI drops well below the 90% mark, and the transport through the QEI is essentially in free-drift, a limit that is reached when sea ice interactions are small and whose magnitude depends chiefly on the simulated wind forcing of the model. We conclude that the enhanced transport regime from 2040-2080 is in near free-drift regime, and therefore realistic.

Fig. S2. Observed and simulated December-January-February (dashed lines) and August-September-October (full lines) mean sea ice concentration for CESM1.3-HR (blue) and CDR (black) in the LIA-N (a) and the QEI (b). The simulated monthly mean time series is shown in light blue.

The revised manuscript now reads:

- In the Introduction: “Also, sea ice velocities are a function of the ice-ice interactions (i.e. the rheology) to the first order, which decreases exponentially with sea ice concentration²⁵.”
- In the Results - 2.1 : “The Arctic-wide thinner ice cover feeding the LIA leads to a somewhat earlier onset of the decline in September sea ice extent and August-September-October mean sea ice concentration in the LIA-N and the QEI simulated by CESM1.3-HR compared to observations (Fig. 2 and S2). ... The simulated August-September-October mean sea ice concentration in the LIA-N and the QEI has mean negative biases of 20% and 9% respectively, compared to CDR (Fig. S2). While there are negative biases in sea ice extent, SIA and sea ice thickness in CESM1.3-HR, the sea ice concentration in the LIA-N is similar to that of CESM1.3-LR (not shown), because of the omni-present sea ice convergence against the northern CAA coastline.

Since sea ice resistance is mostly dependent on ice concentration²⁵, thicker sea ice would lead to similar sea ice area fluxes in the LIA.”

- In the Results - 2.2 : “From 2020 to 2080, the August-September-October mean sea ice concentration in the QEI drops well below 90%, leading to low ice-ice interactions (Fig. S2). In the LIA-N, the trend in August-September-October mean sea ice concentration reverses around 2035, coinciding with consistent pan-Arctic ice-free conditions. With sea ice concentration in the LIA-N remaining between 90% and 40% and allowing year-long ice intake into the QEI, we argue that the simulated sea ice velocities at gates of the QEI are in near free-drift during the 2040-2080 period. This regime of ice transport is reached when ice-ice interactions are small and whose magnitude depends chiefly on the simulated wind forcings of the model. As Arctic wind speeds are projected to increase by 6.4% to 9.6% by the end of the century³⁵, surface wind stress is expected to rise by ~13-20%.”
- In the Online methods: “Section 4.8 Sea ice extent, thickness and concentration ... The mean sea ice concentration time series are computed for grid cells with at least 15% sea ice concentration.”

All differences between the high- and low-resolution configuration of the CESM1.3 are now stated, including their potential link with the biases in the simulations. The revised Introduction now reads: “CESM1.3-HR shows significant negative biases in sea ice extent, area, and thickness when compared to observations and lower-resolution models. These biases are likely linked to a warmer sea surface temperature, which may result from the explicit parameterization of upward ocean heat transport by mesoscale and sub-mesoscale eddies²⁷. This positive bias in the vertical temperature profile leads to a stronger meridional heat transport, a 25% stronger pan-Arctic amplification and consequently an underestimation of the SIE, SIA and SIT in the Northern Hemisphere²⁷. ”

We also provide details on CESM1.3-LR biases in the methods. The end of section 4.2 now reads: “CESM1.3-LR has a significant cold-bias in global surface temperature originating from an implicit parametrization of vertical heat transport.²⁸ A detailed comparison between CESM1.3-HR and CESM1.3-LR and the effects of increased spatial resolution on output diagnostics can be found in Chang et al.²⁸.”

-Related to the above comment and the known issues with the HR run... the HR run has lower sea ice area (SIA) and thickness compared to the low-resolution (LR) run, and there is therefore less ice in the ‘ice shed’ sources to supply the ice to the LIA (i.e. along the marginal shelf seas including the Siberian shelf). As a result one could expect less ridging and thinner ice in the LIA, which will result in more mobile ice, and thus more export through the straits. Further discussion of how less and thinner sea ice in the pan-Arctic could impact results in the LIA would be beneficial.

#4. While there is a bias low in pan-Arctic SIA in CESM1.3-HR, the sea ice concentration in the LIA-N in both high- and low-resolution models are similar because of the omni-present sea ice convergence against the CAA coastline – see Figure in response **#2**, where the mean sea

ice concentration in the LIA-N is shown for both model versions. Since sea ice resistance is mostly dependent on sea ice concentration (Hibler, 1979), thicker sea ice would lead to similar sea ice area fluxes in the LIA. This was clarified in the revised manuscript with the mention “results not shown”.

The revised result section 2.1 now reads: “While there are negative biases in sea ice extent, SIA and sea ice thickness in CESM1.3-HR, the sea ice concentration in the LIA-N is similar to that of CESM1.3-LR (not shown), because of the omni-present sea ice convergence against the northern CAA coastline. Since sea ice resistance is mostly dependent on ice concentration²⁵, thicker sea ice would lead to similar sea ice area fluxes in the LIA.”

-I think the paper would benefit from clearly providing the author’s definition of the Last Ice Area (LIA) quite early in the manuscript because their definition differs somewhat from others used in previous literature, so this could be a source of confusion for readers (specifically that the authors definition includes the CAA, which is not included in some other studies)

#5. In Newton et al., 2021, only the part of the LIA that was resolved in the Polar Pathfinder sea ice drift dataset was included – because this paper focussed on the ice shed of the LIA –, contrary to the definition by the World Wildlife Fund that coined the term LIA following work from some of our co-authors on The Last Ice Refuge. This is causing confusion. The definition of the LIA now includes the Nansen Sound, Jones Sounds, Lincoln Sea and the North Water Polynya, as per the definition from WWF. Note that the southern part of the CAA is not included in the LIA because it is nearly always open in the 21st century. The CAA-South is still discussed in the context of the drain-trap mechanism (i.e. back pressure limiting the flow of ice from CAA-North to CAA-South. This is now clarified, including the mention of the UNESCO ecoregions and Tuvaijuittuq when the LIA is first introduced at the beginning of the Introduction as suggested. Fig. 1 is updated as suggested (see response **#22** regarding Fig.1 for further details).

The Introduction now reads: “Within a few decades, the Arctic Ocean is expected to become seasonally ice-free with the exception of the Queen Elizabeth Islands (QEI) and the region north of the Canadian Arctic Archipelago (CAA) and north of Greenland^{1,2}. This region was first introduced at a press conference at the American Geophysical Union Annual meeting^{3,4} and served as the basis for the Last Ice Area (LIA) flagship project of the World Wildlife Fund - Canada (wwf.ca). In 2019, this led to the interim creation of the Tuvaijuittuq Marine Protected Area within the LIA designated by Canadian ministerial order (Tuvaijuittuq means “the ice never melts” in Inuktitut). The LIA also encompasses the Remnant Arctic Multi-Year Sea Ice and the Northeast Water Polynya ecoregion, which is recognized as a potential UNESCO World Heritage site⁵. The stability of this region is crucial for preserving the Arctic ecology as it provides a suitable habitat for ice-dependent and ice-obligate species, including polar bears, belugas, bowhead whales, walrus, ringed seals, bearded seals and ivory gulls^{6,7}. These Arctic wildlife are a cornerstone of Inuit hunting practices, a heritage now imperiled by the decline of sea ice. In August 2024, interim protected status was extended to the Tuvaijuittuq Marine Protected Area within the LIA for up to five years “while the Government of Canada works with partners to consider long-term protection” (www.dfo-mpa.gc.ca). This analysis provides

foundational information for understanding future sea ice conditions in this area. ... The boundaries of the LIA vary in the literature depending on the long-lasting sea ice cover contour in model projections^{9,16}. In this study, we adopt the definition provided by the WWF which includes the LIA-North (LIA-N), the QEI, and the northern Baffin Bay (see Fig. 1). ... Although the WWF definition of the LIA includes sea ice in northern Baffin Bay and Jones Sound, these areas are currently seasonally ice-free and do not contribute to the LIA (Fig 1).”

-I feel the paper would benefit from providing more results for sea ice trends across the pan-Arctic in the HR run for context for the LIA results. Specifically, one of the main conclusions is that the LIA will disappear one decade after an ice-free pan-Arctic is reached, but the figures in the paper do not make it straight-forward to evaluate this conclusion. The authors could consider moving some of the pan-Arctic data presented in Fig. S5 into the main paper and into some of the related figures (discussed below) to better support this conclusion and provide direct evidence to allow readers to assess this conclusion.

#6. The pan-Arctic data from Fig S5 is now included in the revised Fig 2, 3 and 4 as suggested by the reviewer. We modified the Introduction to introduce known biases, switched sections 2.1 and 2.2 to provide more context on pan-Arctic results before interpreting projections for the LIA. Note that the order of the figures changed following edits of the manuscript. Biases of the model and estimates of ice-free Arctic are now introduced into the Introduction.

The revised Introduction now reads: “CESM1.3-HR shows significant negative biases in sea ice extent, area, and thickness when compared to observations and lower-resolution models. These biases are likely linked to a warmer sea surface temperature, which may result from the explicit parameterization of upward ocean heat transport by mesoscale and sub-mesoscale eddies²⁷. This positive bias in the vertical temperature profile leads to a stronger meridional heat transport, a 25% stronger pan-Arctic amplification and consequently an underestimation of the SIE, SIA and SIT in the Northern Hemisphere²⁷. ... Previous studies using climate models have dedicated considerable effort into estimating a realistic timing of a seasonally ice-free Arctic, by trying to reduce major uncertainties rising from the model, climate variability and the uncertainty in future greenhouse gas scenarios. Most CMIP3 and CMIP5 models failed to reach ice-free conditions at the end of the century, decoupling from the rapid observed decline of pan-Arctic sea ice^{31,32}. The CMIP6 generation of models showed improvements in reproducing observed sea ice conditions^{1,2}. CMIP6 models predict that the central Arctic could experience its first ice-free summer as early as the 2020s or 2030s, with a high likelihood of this occurring by 2050. Under all warming scenarios, the central Arctic Ocean is expected to be ice-free in most summers, including natural variability, by mid-century, around 2035 and 2067¹. Only under the lowest warming scenario (SSP1-1.9) do CMIP6 models keep significant thick September sea ice north of the CAA at the end of the century¹. This thick sea ice disappears 10 years to 20 years after reaching a consistent seasonally ice-free central Arctic under moderate to high warming scenarios¹. As part of the CMIP6 suite, HighResMIP models that resolve the channels of the CAA and Nares Strait generally overestimate SIA loss in the LIA compared to observations, projecting both regional and pan-Arctic seasonally ice-free conditions by around 2050³³. Uncertainties regarding projections for regional ice-free

conditions using CMIP models are greater than those for pan-Arctic projections and are highly sensitive to the specific models employed³⁴. Rather than attempting to project the exact timing of the regional disappearance of the LIA, we provide the fastest rate that it could take to drain and melt the LIA once the central Arctic Ocean becomes seasonally ice-free, thereby accounting for the biases and uncertainties in CESM1.3-HR.”

The Results section 2.2 now starts with: “Within the context of this high-end forcing scenario, the central Arctic Ocean becomes seasonally ice-free in 2020 for the first time, and then continuously from 2035 (Fig. 2). Due to the pan-Arctic biases in sea ice extent and thickness in CESM1.3-HR, these projections fall within the early range of seasonally ice-free pan-Arctic estimates from CMIP6 models¹.”

-Is there opportunity to make more comparisons with other high-resolution sea ice / climate model runs? I personally am not familiar with the full suite of existing high resolution GCMs, but are there not some other high-resolution model runs to compare to (even regional models) to better evaluate the sea ice trends? Perhaps GFDL CM2.6 and HighResMIP CMIP6? If the authors can compare their results with other high resolution model runs this would go a long way toward strengthening their conclusions. And if not, then perhaps more emphasis in the conclusions on what further analysis of this HR run is needed, and what other HR model runs are needed to better evaluate this results (beyond just low-emission scenario runs) would be beneficial.

#7. Other high-resolution projects include the GFDL CM2.6 and HighResMIP. GFDL CM2.6 resolves channels of the CAA and Nares Strait, but has a significant cold bias and remains ice covered (sea ice extent typical of the last decade in the historical record) even after more than doubling of CO₂ at the end of the century (Decuyperè et al. 2022). The HighResMIP models generally reach a seasonally ice-free Arctic around 2050, showing a stronger rate of ice loss compared to observations during the historical period. In the LIA, most HighResMIP models generally overestimate SIA loss compared to observations. When model selection is applied, these biases result in regional ice-free conditions occurring simultaneously with pan-Arctic ice loss. When comparing the timing of pan-Arctic and regional ice-free conditions between HighResMIP and their lower resolution equivalent, Selivanova et al. (2024) found “no clear link between the model resolution and the pace of sea ice loss”. We agree that future studies with higher resolution models would be needed to gain insights on the effects of increased resolution on projections of the LIA. Here, we focus on CESM1.3-HR knowing other models are also biased. We have revised the Introduction to include higher resolution studies insights and have revised the Discussion to include future needs.

The Discussion now reads: “Further research is also needed to identify the factors influencing sea ice trends in the current CESM1.3-HR simulation, along with insights from various high-resolution climate models, to better understand how resolution affects feedbacks and sea ice processes, and to provide a more accurate prognosis for the LIA.”

The Introduction now reads: “As part of the CMIP6 suite, HighResMIP models that resolve the channels of the CAA and Nares Strait generally overestimate SIA loss in the LIA compared to observations, projecting both regional and pan-Arctic seasonally ice-free conditions by around 2050³³. Uncertainties regarding projections for regional ice-free conditions using CMIP models are greater than those for pan-Arctic projections and are highly sensitive to the specific models employed³⁴. Rather than attempting to project the exact timing of the regional disappearance of the LIA, we provide the fastest rate that it could take to drain and melt the LIA once the central Arctic Ocean becomes seasonally ice-free, thereby accounting for the biases and uncertainties in CESM1.3-HR.”

Specific comments:

13-14: here as phrased, the LIA seems to be the area north of Greenland AND north of CAA. This should be revised to make it explicit the authors are referring to LIA as including the whole CAA

#8. This was clarified as suggested by the reviewer. Note that the southern part of the CAA is not included in the LIA because it is nearly always open in the 21st century. The CAA-South is still discussed in the context of the drain-trap mechanism (i.e. back pressure limiting the flow of ice from CAA-North to CAA-South. Please see response **#5** for further details on the definition of the LIA, as well as the revised Fig. 1.

The Introduction now reads: “Within a few decades, the Arctic Ocean is expected to become seasonally ice-free with the exception of the Queen Elizabeth Islands (QEI) and the region north of the Canadian Arctic Archipelago (CAA) and north of Greenland^{1,2}. ”

20-21 - the statement ‘the LIA could disappear a little more than one decade after the Arctic Ocean has reached seasonally ice-free conditions’ is presented as one of the main conclusions, but I found this conclusion difficult to assess with the figures present. See below for suggestions.

#9. As suggested by the reviewer below, Fig S5 is now included into the revised Fig 2,3 and 4, and discussed in the main text making this assessment more direct. We also added more references to figures in the results and more analysis in the Discussion to convey clearer analysis for this key message. A review of Arctic ice-free conditions from various models is now added in the Introduction to compare with the model in the Results 2.2.

The main finding of the manuscript - that the LIA could disappear a little more than one decade after the central Arctic Ocean has reached seasonally ice-free conditions - is only possible to draw when considering the amount of sea ice being fed into the LIA. Those were taken from lower resolution CMIP6 simulations to strengthen the estimate of the timescale required to drain and melt the LIA. The revised Discussion now reads: “Historically, $150 \times 10^3 \text{ km}^2$ of the central Arctic sea ice is transported annually into the LIA-N by the Beaufort Gyre increasing to a maximum of $\sim 1,200 \times 10^3 \text{ km}^2$ annually by the end of the century in a lower emission forcing scenario and nearly zero in high-end scenarios as it melts before reaching the LIA-N⁹.

Across all scenarios, CMIP6 models confirm that this sea ice will be thin and seasonal¹. Reduced sea ice transport through Fram Strait (Fig. 5g), and increased melting from warmer temperatures, indicate that thin pan-Arctic sea ice will not be enough to maintain a perennial ice cover in the LIA, given the fluxes through the QEI and Nares Strait. Consequently, this enhanced ice transport through the channels of the CAA and Nares Strait could lead to the drainage of the LIA in as quickly as 6 years following continuous seasonally ice-free conditions in the central Arctic Ocean, and upon ice being sufficiently thin to allow unobstructed transport.

30 - Tuvaijuittuq should be delineated on Fig 1

10. Corrected as suggested by the reviewer.

34-35 - “These Arctic wildlife are a cornerstone of Inuit hunting practices, a heritage now imperilled by the decline of sea ice.” - true, but it depends on how the authors are defining the LIA and whether it includes the QEI and CAA-S. For the LIA-N alone I am not aware of any Inuit that regularly hunt in that region, so it is not clear how the loss of the ice in the LIA-N will imperil Inuit heritage (one can certainly make indirect connections here). Perhaps this statement could be generalized to discuss pan-Arctic sea ice loss and environmental changes in general, or alternatively focused to the specific impacts of sea ice loss in the LIA (ecological changes, travel/shipping hazards, and impacts to Inuit)

11. The sentence was removed.

36 - I would suggest a clear statement of how the authors are defining the LIA - specifically that it includes the CAA which is in contrast to a number of other definitions of the LIA in the literature - is needed here early in the introduction to avoid confusion and make it clear why the authors are including the QEI and CAA in their discussion. The specific definition of the 3 regions (LIA-N, QEI, and CAA-S) that comes at the end of the introduction can still be included where it is (Lines 75-78), but a more explicit definition of the LIA as authors define it is needed earlier (including mention of what marine conservation areas are included (e.g. includes Tuvaijuittuq, what about Pikialasorsuaq/Sarvarjuuaq? Tallurtiup Imanga?...and each of these should be included in Fig 1 if mentioned in text).

#12. In Newton et al., 2021, only the part of the LIA that was resolved in the Polar Pathfinder sea ice drift dataset was included – because this paper focussed on the ice shed of the LIA – , contrary to the definition by the World Wildlife Fund that coined the term LIA following work from some of our co-authors on The Last Ice Refuge. This is causing confusion. The definition of the LIA now includes the Nansen Sound, Jones Sounds, Lincoln Sea and the North Water Polynya, as per the definition from WWF. Note that the southern part of the CAA is not included in the LIA because it is nearly always open in the 21st century. The CAA-South is still discussed in the context of the drain-trap mechanism (i.e. back pressure limiting the flow of ice from CAA-North to CAA-South. This is now clarified, including the mention of the unesco ecoregions and Tuvaijuittuq when the LIA is first introduced at the beginning of the Introduction as suggested. Fig. 1 is updated as suggested (see response **#22** regarding Fig.1 for further details).

The Introduction now reads: “Within a few decades, the Arctic Ocean is expected to become seasonally ice-free with the exception of the Queen Elizabeth Islands (QEI) and the region north of the Canadian Arctic Archipelago (CAA) and north of Greenland^{1,2}. This region was first introduced at a press conference at the American Geophysical Union Annual meeting^{3,4} and served as the basis for the Last Ice Area (LIA) flagship project of the World Wildlife Fund - Canada (wwf.ca). In 2019, this led to the interim creation of the Tuvaijuittuq Marine Protected Area within the LIA designated by Canadian ministerial order (Tuvaijuittuq means “the ice never melts” in Inuktitut). The LIA also encompasses the Remnant Arctic Multi-Year Sea Ice and the Northeast Water Polynya ecoregion, which is recognized as a potential UNESCO World Heritage site⁵. The stability of this region is crucial for preserving the Arctic ecology as it provides a suitable habitat for ice-dependent and ice-obligate species, including polar bears, belugas, bowhead whales, walruses, ringed seals, bearded seals and ivory gulls^{6,7}. These Arctic wildlife are a cornerstone of Inuit hunting practices, a heritage now imperiled by the decline of sea ice. In August 2024, interim protected status was extended to the Tuvaijuittuq Marine Protected Area within the LIA for up to five years “while the Government of Canada works with partners to consider long-term protection” (www.dfo-mpa.qc.ca). This analysis provides foundational information for understanding future sea ice conditions in this area. ... The boundaries of the LIA vary in the literature depending on the long-lasting sea ice cover contour in model projections^{9,16}. In this study, we adopt the definition provided by the WWF which includes the LIA-North (LIA-N), the QEI, and the northern Baffin Bay (see Fig. 1). ... Although the WWF definition of the LIA includes sea ice in northern Baffin Bay and Jones Sound, these areas are currently seasonally ice-free and do not contribute to the LIA (Fig 1).”

85-92 - these sentences are not results, they are more a summary of methodological choices and I think would be better moved to the last paragraph of the introduction.

The results section would benefit from beginning with a brief description of pan-Arctic sea ice trends in the HR simulation, then transition into specifics about the LIA. This would help provide context for the LIA results, and also better support one of the main conclusions the authors are making related to LIA becoming ice free within one decade after ice-free Arctic.

Related to this, I would also suggest that the authors could include more discussion of the known biases and limitations of the CESM-HR simulation as it relates to Arctic sea ice and temperatures in the introduction. Provide enough information up-front so the reader can evaluate the results in this context on the first read through, and it would allow the authors to pre-emptively justify the HR results and some of their limitations. I think this information could be placed at the end of the introduction.

#13. Line 85-92 are now at the end of the Introduction as suggested. We modified the Introduction to introduce known biases, switched sections 2.1 and 2.2 to provide more context on pan-Arctic results before interpreting projections for the LIA. Note that the order of the figures changed following edits of the manuscript. Biases of the model and estimates of ice-free Arctic are now introduced into the Introduction.

The Introduction now gives a more detailed review of previous projections and uncertainties in the timing of an “ice-free” Arctic – i.e. defined as the time when the multi-model mean sea ice extent reaches one million km² and variability around the mean – as well as the projected disappearance of the Last Ice Area (not formerly reported) from CMIP3, CMIP5 and CMIP6 (Stroeve et al. 2007;2012; Notz 2020; Jahn et al. 2024).

The Introduction now reads: “CESM1.3-HR shows significant negative biases in sea ice extent, area, and thickness when compared to observations and lower-resolution models. These biases are likely linked to a warmer sea surface temperature, which may result from the explicit parameterization of upward ocean heat transport by mesoscale and sub-mesoscale eddies²⁷. This positive bias in the vertical temperature profile leads to a stronger meridional heat transport, a 25% stronger pan-Arctic amplification and consequently an underestimation of the SIE, SIA and SIT in the Northern Hemisphere²⁷.

Previous studies using climate models have dedicated considerable effort into estimating a realistic timing of a seasonally ice-free Arctic, by trying to reduce major uncertainties rising from the model, climate variability and the uncertainty in future greenhouse gas scenarios. Most CMIP3 and CMIP5 models failed to reach ice-free conditions at the end of the century, decoupling from the rapid observed decline of pan-Arctic sea ice^{31,32}. The CMIP6 generation of models showed improvements in reproducing observed sea ice conditions^{1,2}. CMIP6 models predict that the central Arctic could experience its first ice-free summer as early as the 2020s or 2030s, with a high likelihood of this occurring by 2050. Under all warming scenarios, the central Arctic Ocean is expected to be ice-free in most summers, including natural variability, by mid-century, around 2035 and 2067¹. Only under the lowest warming scenario (SSP1-1.9) do CMIP6 models keep significant thick September sea ice north of the CAA at the end of the century¹. This thick sea ice disappears 10 years to 20 years after reaching a consistent seasonally ice-free central Arctic under moderate to high warming scenarios¹. As part of the CMIP6 suite, HighResMIP models that resolve the channels of the CAA and Nares Strait generally overestimate SIA loss in the LIA compared to observations, projecting both regional and pan-Arctic seasonally ice-free conditions by around 2050³³. Uncertainties regarding projections for regional ice-free conditions using CMIP models are greater than those for pan-Arctic projections and are highly sensitive to the specific models employed³⁴. Rather than attempting to project the exact timing of the regional disappearance of the LIA, we provide the fastest rate that it could take to drain and melt the LIA once the central Arctic Ocean becomes seasonally ice-free, thereby accounting for the biases and uncertainties in CESM1.3-HR.”

The Results section 2.2 now starts with: “Within the context of this high-end forcing scenario, the central Arctic Ocean becomes seasonally ice-free in 2020 for the first time, and then continuously from 2035 (Fig. 2). Due to the pan-Arctic biases in sea ice extent and thickness in CESM1.3-HR, these projections fall within the early range of seasonally ice-free pan-Arctic estimates from CMIP6 models¹.”

94-96 - how much of the sea ice loss being dominated by thermodynamic term later in the simulations could be explained by the fact that the CESM-HR is known to run hot in the Arctic?

It appears there is more than just higher spatial resolution at play here that could be affecting the results.

#14. Given that the sea ice motion in the 21st century is mainly in free drift (see response **#3** above), the positive feedback from the thermodynamic (CESM1.3-HR running hot) enhancing the dynamic (via the sea ice concentration dependence of the rheology term resulting in free drift) is not active. In addition, the estimated timescale to eliminate the LIA-N reported in the manuscript is an upper bound estimate based on the dynamical sink term of fluxes through the QEI and Nares Strait, and does not explicitly consider the (biased) thermodynamic sea ice loss.

96-98 - sentence a bit confusing, maybe re-word.

#15. This sentence was clarified as suggested by the reviewer. The new sentence now reads: “The LIA-N and the QEI are regions of thicker sea ice, and therefore show smaller area loss for a reduced ice thickness compared to the CAA-S. In the latter, SIA loss remains thermodynamically driven until near the end of the century.”

98-102 - again, bit hard to follow. Suggest separating concepts out into separate sentences. Focus on LIA-N for first sentence. Focus on QEI for second sentence.

#16. The sentence was modified as suggested by the reviewer. The text now reads: “During the 2040-2080 time period, the LIA-N is mostly seasonally ice-free with a predominance of the 0.64-2.47 m category in May ice thickness distribution (Fig. 3). The loss of SIA is thermodynamically driven in the QEI (Fig. 4 and S4), with no net dynamic loss (flux in = flux out, Fig. 6), but a significant increase in the flux throughflow magnitude serving to drain the seasonal thin cover and the remnant thick ice of the LIA-N (Fig. 3).”

178-180 - Reads “This, coupled with enhanced melting, results in the complete drainage of the LIA and a seasonally ice-free Arctic, until sea ice availability in the source regions (i.e the LIA-N and central Arctic) declines (2081-2100)” but other than Fig. S5, there is no citation and little evidence given to support this statement with respect to a seasonally ice-free Arctic or decline of sea-ice in source regions. I feel perhaps the pan-Arctic data in Fig. S5 could be moved into the main paper and included in Fig. 2, 3, & 6 as separate panel. Note this might also necessitate revising Fig 1 to include a pan-Arctic map and labels (perhaps as a small inset)

#17. The pan-Arctic data from Fig S5 is now included in the revised Fig 2, 3 and 4 as suggested by the reviewer. Fig. 1 now includes a pan-Arctic map as a small inset as suggested. Note that the order of the figures changed following edits of the manuscript. See response **#6** for details on changes regarding seasonally ice-free Arctic assessment.

This passage was clarified and now reads “This, coupled with enhanced melting, and a reduction in sea ice availability in the source regions (i.e. the LIA-N and central Arctic - Fig. 2 and 4), results in the complete drainage of the LIA (Fig. 2). Finally, SIA fluxes through the

QEI reduce during the 2081-2100 period, as winter sea ice declines in the LIA (Fig. 4) and the central Arctic (not shown).”

201-202 - “The fate of the LIA depends chiefly on sea ice conditions north of the CAA, which is the region’s ice source.“ - this statement is somewhat confusing as the region north of the CAA is within the LIA as per Fig 1. And the introduction says the source is actually the marginal shelf seas - please correct this apparent contradiction

#18. Thanks for pointing out this inconsistency. The revised sentence now reads: “The fate of the LIA as a whole depends chiefly on sea ice conditions in its northern part, which hinders sea ice transport and allows replenishment of thick sea ice in the QEI. ”

[1]210–212 - Although running HR simulations under lower emission scenarios would indeed be valuable, I would suggest that further research into what factors are affecting sea ice trends in the existing HR simulation would also be beneficial and informative.

#19. This is now included in the future work section of the revised manuscript as suggested by the reviewer. The end of the Discussion now reads “Further research is also needed to identify the factors influencing sea ice trends in the current CESM1.3-HR simulation, along with insights from various high-resolution climate models, to better understand how resolution affects feedbacks and sea ice processes, and to provide a more accurate prognosis for the LIA.”

215-218 - I feel this statement “This analysis suggests that rapid reductions of emissions of greenhouse gases and carbon capture would be needed for the LIA to function as a bridge to the long-term survival of ice-obligate and ice-dependent species.“ is not well supported by the results presented in this paper. While the authors make inferences about what lower emission scenarios would likely show, those results are not yet available to make this conclusion here. I am also not convinced, given the known biases in the HR model, that all the sea ice loss trends presented in this paper are robust enough to neglect the possibility that some of what has contributed to rapid sea ice loss from the LIA in the HR version could be due to these inherent biases (as opposed to simply being able to resolve the narrow straits of CAA and Nares). I suggest the authors temper their conclusion with these limitations in mind (this is still a valuable paper, and should rightfully stimulate more research into the behavior of sea ice in the HR models and the need to improve them).

#20. The last paragraph of the Discussion was rewritten, to remove conclusions regarding reduction of emissions of greenhouse gasses and carbon capture as suggested by the reviewer. The last sentence of the Discussion, regarding the Tuvaijuituq Marine protected Area was moved in the Introduction.

The end of the Discussion now reads: “The LIA has been envisioned as a refugium during the period of ‘overshoot’ between the current warming and ice melting and when we are able to stabilize temperatures and even trend back toward historical conditions⁹. While there are limitations to this analysis in terms of model biases, findings suggest that for the LIA to function as a bridge to the long-term survival of ice-obligate and ice-dependent species, actions

are needed to reduce warming such that the Arctic Ocean maintains some perennial thick sea ice cover.”

369- Please explicitly explain how sea ice ‘ridging’ term is defined and how it contributes to sea ice loss in the dynamic component

#21. Changes in sea ice concentration can occur because of advection or thermodynamic processes. When the sea ice concentration exceeds 100% due to dynamical processes, it is capped at 100% and the excess sea ice concentration is stored/interpreted as “ridging”. A clear definition is now included in section 4.10 of the revised manuscript: “The area tendency is decomposed into advection and ridging terms, the latter redistributing the thickness distribution allowing SIA to maintain or decrease under dynamical processes.”

Fig. 1 - Three regions are defined (LIA-N, QEI, and CAA-S). My understanding is that the authors intend for all of these three regions combined to form the overarching Last Ice Area. This seems a somewhat different definition than previous literature (e.g Newton et al, 2021), where LIA-N is closer to the standard definition, where there is overlap with Tuvaijuittuq MPA, and excludes the CAA owing to coarse resolution of LR climate models and satellite imagery that cannot resolve narrow passages. Some clarification here is required. I would suggest some overarching outline or shading could be used to delineate the entire LIA (and labelled as such), with the 3 sub-regions also defined as at present.

It is also not clear why certain waterways, including the Nansen and Greely Sounds system, Jones Sound, and coastal waters along N Ellesmere and Greenland and the Lincoln Sea are not included in the shaded areas.

Is it worth including the outline of the ‘Remnant Arctic Multi-Year Sea Ice and the Northeast Water Polynya Ecoregion’ as a site for potential Outstanding Universal Value for inclusion on the UNESCO World Heritage List = Remnant Arctic Multi-Year Sea Ice and the Northeast Water Polynya Ecoregion (<https://portals.iucn.org/library/sites/library/files/documents/2017-006.pdf>)? This region seems to coincide with the LIA as defined by the authors in the present study, so could be useful to include.

#22. In Newton et al., 2021, only the part of the LIA that was resolved in the Polar Pathfinder sea ice drift dataset was included – because this paper focussed on the ice shed of the LIA – , contrary to the definition by the World Wildlife Fund that coined the term LIA following work from some of our co-authors on The Last Ice Refuge. The definition of the LIA now includes the Nansen Sound, Jones Sounds, Lincoln Sea and the North Water Polynya, as per the definition from WWF. This is now clarified, including the mention of the potential unesco ecoregion, when the LIA is first introduced at the beginning of the Introduction. Note that this new definition of the LIA does not change the results since the Nansen Sound is not resolved even in the CESM1.3-HR (see section 4.9 Sea ice area fluxes at key gates). Therefore, Nansen and Greely sounds are part of the LIA, but are not included in the QEI region, because thick ice accumulates there and can only melt, leading to unrealistic thickness values. The area of the LIA-N changed from 1.12 to 1.14 million km² by including coastal waters near Greenland

and Ellesmere Island, as suggested by the reviewer. The few grid points that were added did not significantly change the results - all LIA-N panels of Figures were updated with new results.

Fig 1. caption now reads: “Fig. 1. Map of the Last Ice Area (red contour) as defined by the World Wildlife Fund, including the Queen Elizabeth Islands (QEI, yellow), the region north of the Canadian Arctic Archipelago (LIA-N, blue) and part of the CAA-South (orange), with major gates connecting the LIA-N, the QEI, the CAA-S (orange) and the Northern Baffin Bay (white). These include the QEI-In gates [Ballantyne Strait (Ball.), Wilkins Strait, Prince Gustaf Adolf Sea (Pr. GA.), Peary Channel, Sverdrup Channel (Sv.), and Eureka Sound (Eur.)], the QEI-Out gates [Fitzwilliam Strait (Fit.), Byam Martin Channel (BMC.), Penny Strait (Pen.), Cardigan Strait and Hell Gate (H)], Fram Strait, Nares Strait, Jones Sound, Lancaster Sound (Lanc.), Amundsen Gulf (Amund.) and M’Clure Strait. The Tuvaijuittuq Marine Protected Area (in the LIA-N) is hashed and the Remnant Arctic Multi-Year Sea Ice and the Northeast Water Polynya ecoregion is dotted.”

Minor notes:

-A higher resolution image would improve the map quality.

#23. Corrected as suggested by the reviewer. All figures were saved with a higher resolution.

-“Eur.” is shown on map but not defined in caption.

#24. Corrected as suggested by the reviewer.

-The “H” for Hells Gate should be moved a bit to the left closer to the gateway.

#25. Corrected as suggested by the reviewer.

Fig. 4 - bit confusing the time periods of the model simulations vs observation

Also the legend has time periods label (2081, 2100) - does this mean 2081 to 2100? If so perhaps label as 2081-2100?

#26. Corrected as suggested by the reviewer (i.e. 2081-2100).

Plotted are 20-year averages, but the 2001-2020 model results are not 20-year averages, they are broken into smaller periods to match obs. This comparison is great, but it adds some confusion. Perhaps include a full 2001-2020 average to be consistent with other 20-year periods, and then add in an additional line that only plots the shorter observational comparison period? Though I expect this would not add any meaningful difference to the plot, and perhaps just clutter it unnecessarily? I leave it to the authors to determine.

Alternatively, perhaps just state that “shown are 20-year averages, except for the 1921-1980 long-term average, and the 2001-2020 period where a subset of years are averaged to coincide with the observational records as follows...”

#27. Modified the caption as suggested. Caption now reads “Fig. 5. Simulated (full lines) 20-years averaged seasonal cycle of SIA export through the QEI-In (a), QEI-Out (b), Nares Strait (c), M’Clure Strait (d), Lancaster Sound (e), Amundsen Gulf (f) and Fram Strait (g) from 1921 to 2100, except for the 1921-1980 long time average, and the 2001-2020 period where subsets of years are selected to coincide with observational records. The observed (and comparable simulated) mean seasonal cycles are calculated over 2001-2020 for the QEI-In and M’Clure

Strait, 2017-2021 for the QEI-Out, Nares Strait, Lancaster Sound and Amundsen Gulf, and 2000-2014 for the Fram Strait.”

Fig. S2 - please indicate what the solid line and the shaded areas mean? The mean of 2 ensemble members and variance?

-there is overlap in y-axis labels

#28. We believe this comment refers to Fig S1, now Fig. S4. The solid line and shaded area are defined in the caption of the revised Fig. S4 as suggested. Caption now reads: “Fig. S4. Simulated ensemble mean and range dynamic (advection and ridging) and thermodynamic SIA loss integrated spatially and temporally over the melt season for the LIA-N (a), QEI (b) and CAA-S (c).”

The y-axis overlap in Fig S2, now Fig. S5 is removed, as suggested.

-the thermodynamic loss is related to increasing air and sea surface temperatures - but the HR simulation is known to be biased warm in this regard. Further acknowledgement and discussion of this bias and its effect on the results is warranted.

#29. Given that the sea ice motion in the 21st century is mainly in free drift (see response **#3** above), the positive feedback from the thermodynamic (CESM1.3-HR running hot) enhancing the dynamic (via the sea ice concentration dependence of the rheology term resulting in free drift) is not active. In addition, the estimated timescale to eliminate the LIA-N reported in the manuscript is an upper bound estimate based on the dynamical sink term of fluxes through the QEI and Nares Strait, and does not explicitly consider the (biased) thermodynamic sea ice loss. Results are now interpreted in light of an additional analysis on sea ice concentration and supported by a further analysis of biases in the model (See response **#3**).

Fig. S5- CESM-LR does better job with SIA, SIE, only HR does better with thickness compared to observations. This reduces how convincing the sea ice results from the HR run are, but as long as this is clearly addressed in the paper that is sufficient.

Fig. S6 - HR vastly underestimates concentration, and somewhat underestimates thickness. Again, this reduces how convincing the sea ice results from the HR run are, but as long as this is clearly addressed in the text of the paper that is sufficient. A higher resolution figure would help the image quality of this figure.

#30. While there is a bias low in pan-Arctic SIA in CESM1.3-HR, the sea ice concentration in the LIA-N in both high- and low-resolution models are similar because of the omni-present sea ice convergence against the CAA coastline – see Figure in response **#2**, where the mean sea ice concentration in the LIA-N is shown for both model versions. Since sea ice resistance is mostly dependent on sea ice concentration (Hibler, 1979), thicker sea ice would lead to similar sea ice area fluxes in the LIA. This was clarified in the revised manuscript with the mention “results not shown”. Results are now interpreted in light of an additional analysis on sea ice concentration and supported by a further analysis of biases in the model (See response **#3**).

The revised result section 2.1 now reads: “While there are negative biases in sea ice extent, SIA and sea ice thickness in CESM1.3-HR, the sea ice concentration in the LIA-N is similar to that of CESM1.3-LR (not shown), because of the omni-present sea ice convergence against the northern CAA coastline. Since sea ice resistance is mostly dependent on ice concentration²⁵, thicker sea ice would lead to similar sea ice area fluxes in the LIA.”

Please note that Fig. S5 was incorporated into Fig. 2, 3 and 4 and that figure's resolution was increased.

Response to reviewers - Revisiting the Last Ice Area Projections from a High-Resolution Global Earth System Model
Madeleine Fol, Bruno Tremblay, Stephanie Pfirman, Robert Newton, Stephen Howell, Jean-François Lemieux

Dear reviewers,

We want to thank you for a careful review of the manuscript, which has led to a much improved and clearer manuscript. You will find below the point-by-point responses to each of the reviewer's comments in blue and black respectively. We hope that the proposed changes will respond to the reviewers' concerns. New passages are marked in green and deleted passages in red.

Point-by-point revisions from reviewers' comments:

Reviewer #1 :

The authors have done a good job in responding to my concerns by clarifying the main conclusions and by further discussing the potential impact of model biases in CESM-HR on the results. I now recommend publication. This study will be a nice addition to the literature, as it improves our understanding of the changing sea-ice dynamics in the LIA and their impact on the fate of the LIA sea ice, which is of interest for a broad field of scientific research.

Here are some very few minor comments:

L139-140: similar to that of CESM2-LE?

Corrected as suggested by the reviewer.

L214 – 215: I think the increase in mean sea-ice concentration is a bit of an artifact resulting from only selecting grid cells with >15% SIC. As seen in Fig S1, there is essentially no sea ice left in Sept. 2041-60, except for some coastal ice, which is then used to calculate the mean.

This is correct. Thank you for highlighting it. The revised manuscript - section 2.2 now reads: "In the LIA-N, the trend in August-September-October mean sea ice concentration reverses around 2035, when only a few high sea ice concentration grid cells remain (Fig. S1-2)."

L267: As I stated previously, studies 3, 8 and 9 are all based on CCSM/CESM, so I would suggest specifying that here

Corrected as suggested by the reviewer. The revised manuscript now reads: "Earlier studies from low resolution climate models (CCSM or CESM)^{3,8,9} projected that sea ice cover will continue to persist in the LIA several decades past this time."

Reviewer #2 (Remarks to the Author):

General Comments

The authors have addressed most of the reviewer comments thoughtfully, and I thank them for the effort in considering both Reviewer #1 and my own comments. However, I am still left with a few questions:

1) The main conclusion of the study is that the LIA will disappear within one decade after an ice-free Arctic is reached. The metric for an ice free Arctic ($<1,000,000\text{km}^2$) is presented, but what is the metric for 'disappearance' of sea ice in the LIA?

I can only assume the authors mean zero sea ice extent in the LIA? And related to this, although the inclusion of the pan-Arctic data in Fig 2 is an improvement, I still do not think it clearly shows that the LIA will disappear within 1 decade after ice-free Arctic. I am left pondering whether this conclusion is well-supported by the data on-hand. Fig. 4 shows the pan-Arctic dropping to ice-free in 2035, and maybe the LIA-N dropping to zero sometime between 2040-2050, but QEI not until after 2060 (and CAA-S is not part of LIA). So unless the metric is something higher than zero SIA, I am left confused. Perhaps demarcating (vertical or horizontal lines) on these figures when these milestones are reached is needed to clarify the timing? Quite frankly, in my interpretation of Fig. 2, I do not see that the LIA-N, or QEI, or CAA-S reach zero extent within one decade after pan-Arctic drops to ice free conditions. So I feel the authors really need to clearly, and unambiguously, lead the reader to the evidence for this conclusion.

Figure 4 suggests that the QEI is free around 2060 in apparent contradiction with the conclusion stated in the paper (based on characteristic sea ice area fluxes). The scale of the y-axis, however, is one and two orders of magnitude smaller than that of the LIA-N and central Arctic, and the QEI in the middle of the century is practically ice free (one order of magnitude smaller than that of the QEI). For this reason and because the number of ensemble members is still small, we have opted throughout the paper to give an estimate of the timescale needed to drain the LIA from its ice using sea ice fluxes, rather than the projections of sea ice area.

The caption of Figure 2 and 4 now reads: “Note that the vertical scale of each panel is different”.

On a related note, on the metric of $<1,000,000\text{km}^2$ as ice-free Arctic. Presumably the remaining sea-ice that makes up that last $1,000,000\text{km}^2$ is that which is found in the LIA. So in fact the authors are concluding that the LIA will disappear within one decade after the LIA is the only sea ice left in the Arctic. There is some circular reasoning here that is perhaps unnecessary, but maybe an unfortunate artifact of the accepted definition of an ice-free Arctic. It might be prudent to consistently use the term 'ice-free central Arctic' to distinguish the regions being referred to, and help reduce the circularity.

We now use the terms “ice-free central Arctic” and “disappearance of the LIA” to differentiate between the standard 1.0 million km^2 definition from the last IPCC widely used in the community and an Arctic that is fully ice Arctic with no ice left in the LIA, as suggested by the reviewer.

The introduction now reads : “Finally, we discuss the dynamic and thermodynamic contributions, and potential feedback, responsible for the disappearance – i.e., near zero sea ice area - of perennial sea ice of the LIA.”

The introduction was modified accordingly: “Therefore, rather than attempting to project the exact timing of the regional disappearance of the LIA, we provide an estimate of the timescale required to drain-and-melt the LIA once the central Arctic Ocean becomes seasonally ice-free, thereby accounting for the biases and uncertainties in CESM1.3-HR”.

2) The authors have done well to include more discussion of the biases of the model, and have presented more background literature in the introduction. However, in the introduction, when referring to Jahn et al. 2024, the authors state: "This thick sea ice disappears 10 years to 20 years after reaching a consistent

seasonally ice-free central Arctic under moderate to high warming scenarios." To me this seems to indicate that the main conclusion has already been presented in the literature, which appears to reduce the novelty of the current manuscript. The authors very clearly need to state how their work is novel and distinct (in methods or insight) if the basic conclusion is not much different than previous literature.

The paper gives an estimate of the timescale required to drain the LIA dynamically via sea ice transport through the narrow of the archipelago, contrary to Jahn et al (2024) where sea ice transport is not resolved. The introduction of the manuscript now reads: "In moderate to high warming scenarios, the thick sea ice of the QEI disappears thermodynamically 10 to 20 years after reaching a consistent seasonally ice-free central Arctic¹".

This is now also clarified in the discussion of the manuscript: "Therefore, we infer that the LIA could disappear through transport in a little more than one decade (~~2 x 6 years~~) after the central Arctic continuously reaches seasonally ice-free conditions. Note that this conclusion is drawn from the dynamic sea ice mass budget only and is in agreement with recent projections of sea ice extent using CMIP6 models¹. "

3) There remain some confusing structural or grammatical issues in the text. I recommend the authors review the paper to ensure all text is clear and unambiguous, especially the transitions where new sentences have been inserted. There are some minor edits that would help, I have suggested a few below, but please review the full paper for clarity, readability, and flow.

All suggested edits below have been corrected as suggested by the reviewer. The paper was reviewed by all co-authors before resubmission. Some sentences of the discussion have been relocated to improve clarity.

Specific comments

25 - I feel these opening statements need some minor clarification to emphasize the importance and characteristics of the LIA - particularly to highlight that the LIA area is where the oldest, thickest sea ice is, and is expected to persist for some amount of time even after the rest of the Arctic becomes seasonally ice free. Perhaps consider revising this opening sentences to something like: "Within a few decades, most of the Arctic Ocean is expected to become seasonally sea ice-free. The exception to this trend is the region surrounding the QEI and areas north of the CAA and Greenland, where the oldest, thickest sea ice accumulates, and is where perennial sea ice cover is expected to persist (for some time period) even after the remainder of the Arctic is ice free in summer. This region has been called the Last Ice Area (LIA), a concept that was first..."

Some revision along these lines would be beneficial, but please rewrite as the authors see fit.

The opening statements were revised as suggested. The introduction previously read "Within a few decades, the Arctic Ocean is expected to become seasonally ice-free with the exception of the Queen Elizabeth Islands (QEI) and the region north of the Canadian Arctic Archipelago (CAA) and north of Greenland^{1,2}. This region was first introduced at a press conference at the American Geophysical Union Annual meeting^{3,4} and served as the basis for the Last Ice Area (LIA) flagship project of the World Wildlife Fund - Canada (wwf.ca)." and now reads: "Within a few decades, most of the central Arctic Ocean is expected to become seasonally ice-free. The exception to this trend is the region surrounding the Queen Elizabeth Islands (QEI) and areas north of the Canadian Arctic Archipelago (CAA) and Greenland^{1,2}, where some of the oldest and thickest sea ice accumulates. Here, a perennial sea ice cover is expected to persist for some time period even after the remainder of the central Arctic Ocean is ice-

free in summer. This region is termed the Last Ice Area (LIA – see red contour on Fig. 1), a concept that was first introduced at a press conference at the American Geophysical Union Annual meeting^{3,4} and served as the basis for the LIA flagship project of the World Wildlife Fund - Canada (wwf.ca)”.

27 - 'This region was first..' = the region was already there, it is more the concept of the LIA that was introduced at AGU (which is how it was stated in the original submission) - but the phrase 'Last Ice Area' needs to be presented prior to stating that the concept was introduced, as above.

This was corrected as suggested. The introduction now reads: “This region is termed the Last Ice Area (LIA – see red contour on Fig. 1), a concept that was first introduced at a press conference at the American Geophysical Union Annual meeting^{3,4} and served as the basis for the LIA flagship project of the World Wildlife Fund - Canada (wwf.ca).”.

38 - 'This analysis provides...' - some uncertainty which analysis the authors are referring to, presumably the present study, but this sentence would benefit from some clarification.

This passage was corrected. The introduction now reads: “The results presented in this study provide foundational information for understanding future sea ice conditions in this area.”

50-52 - these two new sentences are somewhat out of place as the opener to a paragraph that is mainly about sea ice circulation. Would suggest moving these sentence into paragraph starting at Line 84, perhaps somewhere around Line 86.

The two sentences were moved to Line 86, including a reference to the red contour on Fig. 1.

52 - 'and the northern Baffin Bay.' - remove 'the'. Corrected here and in the caption of Figure 1, as suggested by the reviewers.

55-59 - four advective pathways are listed, but there is no mention of possibility of thermodynamic loss in the LIA. If this paragraph is focus on dynamic processes then please state as much in the paragraph opening, otherwise consideration of thermodynamic process is warranted.

This is corrected as suggested. The first sentence of the paragraph now reads: “The existence of the LIA is primarily due to sea ice dynamic processes, specifically the convergence of sea ice on the northern shores of Canada and Greenland...”.

And regardless of whether it is presented here or elsewhere, I feel that further discussion of factors influencing thermodynamic sea ice loss is needed, including splitting out atmospheric versus oceanic components. How much thermodynamic loss is due to surface melting versus basal melting? Can this tell us anything about whether it is more important for the climate models to get the atmospheric warming right, or the sea surface temperature/salinity right?

We have now added a figure in the supplemental material showing the basal, surface and lateral melt integrated over the melt season for the LIA-N, the QEI and the CAA-S. Results show that the basal melt dominates over surface melt in the LIA-N and is of equal importance in the QEI and CAA-S. The lateral melt in all cases is negligible (see Figure S5). Section 2.2 now reads : “Basal melt dominates over surface melt in the LIA-N, and is of equal importance in the QEI and CAA-S (Fig. S5).”.

65-69 - This paragraph seems out of place and incomplete unto itself. Perhaps it can be combined with the paragraph below (70-83)?

This paragraph was combined with the paragraph below as suggested.

80 - with increasing or decreasing sea ice concentrations? (presumably increasing concentrations, but please explicitly state as much)

This was revised as suggested. The sentence now reads: “Also, the departure from free-drift conditions is a function (to first order) of ice-ice interactions (i.e. the rheology), which decreases exponentially with decreasing sea ice concentration.”

95-96 - this sentence needs revising...present and interpret what exactly about the LIA? in context of what biases?

The sentence was revised as “Then, we present and interpret results for sea ice within the LIA in the context of the model biases.”

106-107 - please explicitly state that these are known bias in the literature, and include an appropriate citation

The reference of Chang et al. (2020) is now included.

112 - I feel this paragraph needs a concluding sentence to address how these biases might impact the results and conclusions that will follow, or some statement saying that these biases will be discussed further later in the paper.

We added a sentence as suggested by the reviewer: “Therefore, considering these negative biases and the RCP8.5 warming scenario, the results presented here should be viewed as a worst-case scenario for LIA ice loss.”

122 - 'around 2025 and 2067.' - do the authors mean 'between 2025 and 2067'?

Yes this is correct. Corrected as suggested.

123-125 - this sentence seems to state that the main conclusion of this paper actually already exists in the literature.

We now clarify that previous studies only consider thermodynamic processes. The revised sentence now reads: “In moderate to high warming scenarios, the thick sea ice of the QEI disappears thermodynamically 10 to 20 years after reaching a consistent seasonally ice-free central Arctic.”

240 - until 2000 = is this the correct year? This seems confusing because the previous sentence refers to 2040-2060 time period, and this sentence begins with 'This gradual loss..' indicating that the authors are referring to the 2040-2060 time period.

Thank you for pointing out this inconsistency. The time periods were corrected. The passage now reads: “This gradual loss of the thick ice within the LIA operates through a slow increase in sea ice advection through the QEI and Nares Strait and out of the LIA-N from 2000 until 2040, a period during which the length of the melt season increases, the ice arches are weaker, and SIA loss becomes thermodynamically driven. At this point, sea ice in the thicker ice categories is reduced by a factor of two in the LIA-N and the QEI, leading to nearly continuous (unobstructed) dynamic export of Arctic sea ice through the QEI and Nares Strait from 2040 to 2080.”

246 - 'complete drainage of the LIA.' By when? Add a date (or duration after ice-free Arctic if preferred)

We now refer to the paragraph below where an estimate of the timescale is given based on sea ice area fluxes. “This, coupled with enhanced melting, and a reduction in sea ice availability in the source regions (i.e. the LIA-N and central Arctic - Fig. 2 and 4), results in the complete drainage of the LIA (Fig. 2 - see below for an estimate based on SIA fluxes).”

265 - drainage of the LIA in as quickly as 6 years..' - where are the authors getting this interpretation from? Can they refer to a specific figure? I presume it is from Line 250, the 6 year residence time, but if so please be explicit in this connection as this is 6 years is presented in a different paragraph, so the connection is tenuous. My preference is that the authors can refer to a specific figure (Such as Fig 2) showing ice in the LIA disappearing with 6 years after ice-free pan-Arctic, but right now I fail to see this clearly in any figure.

The 6 year estimate comes from the sea ice area fluxes and residence time analysis presented in the previous paragraph. The discussion was corrected and some sentences of the discussion have been relocated to improve clarity.

The discussion previously read in three separate paragraphs: “The mean residence time of sea ice in the $1,200 \times 10^3 \text{ km}^2$ LIA-N transitions from 12 years in 1920-1980 to 6 years in 2040-2080. The average throughflows exported through the QEI and Nares Strait for these two time-slices are $\sim (20 + 80) \times 10^3 \text{ km}^2/\text{yr}$ and $\sim (100 + 110) \times 10^3 \text{ km}^2/\text{yr}$ respectively (Fig. 6). ... Consequently, this enhanced ice transport through the channels of the CAA and Nares Strait could lead to the drainage of the LIA in as quickly as 6 years following continuous seasonally ice-free conditions in the central Arctic Ocean, and upon ice being sufficiently thin to allow unobstructed transport. ... Our results suggest that the last 1.2 million km^2 can be drained through the channels of the CAA and Nares Strait in 6 to 12 years with SIA fluxes doubled that of today's climate or that of the future. Therefore, we estimate that the LIA could disappear in a little more than one decade (2×6 years) after the central Arctic continuously-reaches seasonally ice-free conditions.”

The discussion now reads: “Our results suggest that the last 1.2 million km^2 can be drained through the channels of the CAA and Nares Strait in 6 to 24 years. The 6 years are derived from the projected SIA fluxes from the 2040-2080 time period. The average throughflows exported through the QEI and Nares Strait for the 1920-1980 and 2040-2080 time periods are $\sim (0.02 + 0.08) \times 10^6 \text{ km}^2/\text{yr}$ and $\sim (0.1 + 0.11) \times 10^6 \text{ km}^2/\text{yr}$ (Fig. 6). This translates to a change of the mean residence time of sea ice in the $1.2 \times 10^6 \text{ km}^2$ LIA-N from 12 years ($= 1.2/0.1 \text{ yrs}$) to 6 years ($= 1.2/0.21 \text{ yrs}$). This reduced residence time of sea ice through the channels of the CAA and Nares Strait could lead to the drainage of the LIA in as quickly as 6 years following continuous seasonally ice-free conditions in the central Arctic Ocean and upon ice being sufficiently thin to allow unobstructed transport (see Fig. 6). The very-conservative 24 years are derived from the observed estimates from the historical (1997-2022) period, which are about half the simulated values for the same period. Therefore, we infer that the LIA could disappear through transport in a little more than one decade (~~2×6 years~~) after the central Arctic continuously reaches seasonally ice-free conditions. Note that this conclusion is drawn from the dynamic sea ice mass budget only and is in agreement with recent projections of sea ice extent using CMIP6 models¹.”

267-273 - the authors estimate disappearance of the LIA in a little over a decade after ice-free Arctic, and state (2×6 years) - but where are they getting the 2×6 years. I think I see where the 6 years is from, but why multiplying by 2? What is the justification?

The paragraph was rephrased for clarity. We now give two separate estimates, one from the simulated sea ice area fluxes from 2040-2060 and one (very conservative) using the observed historical (1997-2022) sea ice fluxes derived from satellite measurements.

The discussion now reads: “Our results suggest that the last 1.2 million km^2 can be drained through the channels of the CAA and Nares Strait in 6 to 24 years. The 6 years are derived from the projected SIA fluxes from the 2040-2080 time period. The average throughflows exported through the QEI and Nares

Strait for the 1920-1980 and 2040-2080 time periods are $\sim (0.02 + 0.08) \times 10^6 \text{ km}^2/\text{yr}$ and $\sim (0.1 + 0.11) \times 10^6 \text{ km}^2/\text{yr}$ (Fig. 6). This translates to a change of the mean residence time of sea ice in the $1.2 \times 10^6 \text{ km}^2$ LIA-N from 12 years ($= 1.2/0.1 \text{ yrs}$) to 6 years ($= 1.2/0.21 \text{ yrs}$). This reduced residence time of sea ice through the channels of the CAA and Nares Strait could lead to the drainage of the LIA in as quickly as 6 years following continuous seasonally ice-free conditions in the central Arctic Ocean and upon ice being sufficiently thin to allow unobstructed transport (see Fig. 6). The very-conservative 24 years are derived from the observed estimates from the historical (1997-2022) period, which are about half the simulated values for the same period. Therefore, we infer that the LIA could disappear through transport in a little more than one decade ~~(2 x 6 years)~~ after the central Arctic continuously reaches seasonally ice-free conditions. Note that this conclusion is drawn from the dynamic sea ice mass budget only and is in agreement with recent projections of sea ice extent using CMIP6 models¹.”

271 - 'or that of the future.' = this seems confusing, how do you double sea ice fluxes of the future?
See response above.

274 - 'and given that' - This sentence needs revising.
We replaced “given that” with “where”.

Fig. 1 - the revisions to Fig 1 are appreciated in how they present the various regions and flux gates. However I think the figure could be improved in two ways:

1) consider using a different projection, or at least a rectangular aspect ratio to the map, in order to make better use of space (right now over half the map is 'wasted' space (i.e. the lower right-hand $\sim 1/3$ of the map is not utilized), and to centre and focus the map on the LIA/QEI/CAA

We removed the first 5 degree latitude in the figure and decided to keep the projection given that Greenland must be included because we are referring to the Fram Strait in the text.

2) consider moving the flux gate labels off to the side and draw thin lines connecting the label to the strait of interest, thus reducing the clutter of labels, the need for text background shading and text box outlines, and the fact that they obscure geographic features

The gate labels were moved to the side and are connected to the appropriate gates by thin lines, as suggested.

Fig. 2 & 4 - I feel some demarcation showing clearly the timing of the disappearance of sea ice in the LIA is needed, to allow the reader to see that this occurs within 1 decade after ice-free central Arctic.

We decided not to include vertical lines indicating ice-free conditions in Figure 2 and 4 since we discuss the timing of the disappearance of the LIA based on sea ice area fluxes as opposed to the (more subjective) projections of sea ice area.